# Interactions between the NLRP3-Dependent IL-1β and the Type I Interferon Pathways in Human Plasmacytoid Dendritic Cells

**DOI:** 10.3390/ijms232012154

**Published:** 2022-10-12

**Authors:** Dóra Bencze, Tünde Fekete, Walter Pfliegler, Árpád Szöőr, Eszter Csoma, Antónia Szántó, Tünde Tarr, Attila Bácsi, Lajos Kemény, Zoltán Veréb, Kitti Pázmándi

**Affiliations:** 1Department of Immunology, Faculty of Medicine, University of Debrecen, 4032 Debrecen, Hungary; 2Doctoral School of Molecular Cell and Immune Biology, University of Debrecen, 4032 Debrecen, Hungary; 3Department of Molecular Biotechnology and Microbiology, Faculty of Science and Technology, University of Debrecen, 4032 Debrecen, Hungary; 4Department of Biophysics and Cell Biology, Faculty of Medicine, University of Debrecen, 4032 Debrecen, Hungary; 5Department of Medical Microbiology, Faculty of Medicine, University of Debrecen, 4032 Debrecen, Hungary; 6Division of Clinical Immunology, Faculty of Medicine, University of Debrecen, 4032 Debrecen, Hungary; 7Regenerative Medicine and Cellular Pharmacology Laboratory, Department of Dermatology and Allergology, Faculty of Medicine, University of Szeged, 6720 Szeged, Hungary

**Keywords:** plasmacytoid dendritic cell, NLRP3, IL-1β, type I interferon, inflammasome, interaction, inhibition, psoriasis

## Abstract

Generally, a reciprocal antagonistic interaction exists between the antiviral type I interferon (IFN) and the antibacterial nucleotide-binding oligomerization domain (NOD)-like receptor pyrin domain containing 3 (NLRP3)-dependent IL-1β pathways that can significantly shape immune responses. Plasmacytoid dendritic cells (pDCs), as professional type I IFN-producing cells, are the major coordinators of antiviral immunity; however, their NLRP3-dependent IL-1β secretory pathway is poorly studied. Our aim was to determine the functional activity of the IL-1β pathway and its possible interaction with the type I IFN pathway in pDCs. We found that potent nuclear factor-kappa B (NF-κB) inducers promote higher levels of pro-IL-1β during priming compared to those activation signals, which mainly trigger interferon regulatory factor (IRF)-mediated type I IFN production. The generation of cleaved IL-1β requires certain secondary signals in pDCs and IFN-α or type I IFN-inducing viruses inhibit IL-1β production of pDCs, presumably by promoting the expression of various NLRP3 pathway inhibitors. In line with that, we detected significantly lower IL-1β production in pDCs of psoriasis patients with elevated IFN-α levels. Collectively, our results show that the NLRP3-dependent IL-1β secretory pathway is inducible in pDCs; however, it may only prevail under inflammatory conditions, in which the type I IFN pathway is not dominant.

## 1. Introduction

Plasmacytoid dendritic cells (pDC) are a rare but essential subset of dendritic cells (DCs) that act as professional type I interferon (IFN) producing cells by effectively inhibiting viral replication and by triggering a strong antiviral state in the host cells. Although they represent only 0.2–0.8% of human peripheral blood mononuclear cells (PBMCs), pDCs produce 200–1000-times more type I IFN than any other white blood cells in response to various viruses and, thus, play a crucial role in antiviral immunity [1]. The robust type I IFN production of pDCs is critical for the control of acute viral infections [2]. However, the overactivity of pDCs and the overproduction of type I IFNs can induce undesirable autoimmune responses, leading to the damage of the body’s own tissues and to the development of autoimmune diseases. Previous studies have shown that pDCs are directly involved in the pathogenesis and pathomechanism of several autoimmune diseases, such as systemic lupus erythematosus (SLE) or psoriasis [3], thus, serving as potential therapeutic targets in these pathologies. Therefore, the type I IFN response of pDCs is tightly regulated, partially through receptor interactions [4], the comprehensive study of which may help to better understand the mechanism of both antiviral responses and autoimmune diseases.

The functionality of pDCs, including their activation and IFN-producing capacity, is mediated through their specific pattern recognition receptors (PRRs). PDCs selectively express the endosomal Toll-like receptor (TLR) 7 and 9 receptors, which are specialized in the recognition of microbial nucleic acids, mainly of viral origin [5], and, thus, are essential to the production of large amounts of type I IFNs [6]. However, pDCs express several other cytosolic sensors and only those that can detect viruses have been well characterized so far. Our research group previously demonstrated that the two major members of the retinoic acid-inducible gene-I (RIG-I)-like receptors (RLR), namely RIG-I and Melanoma Differentiation-Associated gene 5 (MDA5), are also actively functional in human pDCs and involved in the recognition of replicating viruses in the cytosol [7,8]. The cytosolic DNA sensing cyclic GMP-AMP synthase (cGAS)-stimulator of interferon genes (STING) pathway can also be active in pDCs, resulting in type I IFN production [9]. In addition, our research group has also demonstrated that regulatory receptors of the cytosolic nucleotide-binding oligomerization domain (NOD)-like receptor (NLR) family, namely NLR family caspase activation and recruitment domain (CARD) containing 5 (NLRC5) and NLR family member X1 (NLRX1), are also expressed in pDCs and, among others, are involved in the regulation of RLR activity [8]. Nevertheless, the best-characterized function of NLRs, namely their ability to induce inflammasome activation, has not been extensively studied in pDCs yet. Among the NLRs, NLR pyrin domain containing (NLRP) 1, NLRP3, NLRC4 and, in addition to those absent in melanoma 2 (AIM2), interferon-γ inducible protein 16 (IFI16) and pyrin inflammasomes [10] can form multimeric protein complexes that serve as activation platforms for caspase-1, which controls the maturation of the pro-inflammatory cytokines interleukin (IL)-1β and IL-18. Furthermore, several other NLR-family proteins, including NLRP2, NLRP6, NLRP7, NLRP9b and NLRP12, have been proposed to form inflammasome complexes [11,12,13,14,15]. The role of these receptors is extensively characterized in monocytes, macrophages, conventional DCs and in other cell types [16,17,18,19,20] but not in pDCs. So far, among the inflammasome forming receptors, only the AIM2 cytosolic DNA sensor has been found to be active in tumor-associated pDCs [21] indicating that functional inflammasomes can be assembled in pDCs as well.

Among the caspase-1 activating inflammasomes, the NLRP3 inflammasome is the best studied and most characterized. NLRP3 contains an N-terminal pyrin domain (PYD), a central NACHT (NAIP [neuronal apoptosis inhibitory protein]) domain, which is responsible for activation-induced oligomerization through its adenosine triphosphatase (ATPase) activity [22], and a C-terminal leucine-rich repeat (LRR) domain. Canonical NLRP3 activation usually occurs in two steps. First, during the so-called priming phase, cytokine receptors or other PRRs, such as TLRs and NOD receptors, activate the nuclear factor-kappa B (NF-κB) pathway, which induces the expression of NLRP3 inflammasome components and, thus, contributes to the production of the pro-form of IL-1β and IL-18 [23,24,25,26]. The priming also triggers post-translational modifications, which stabilize NLRP3 in a signal-competent but inactive state [27]. The priming phase is followed by the second NLRP3 activation step, which initiates NLRP3 inflammasome oligomerization. This step can be induced by a wide range of factors, including cellular stress signals initiated during the priming phase and by various pathogen-associated molecular patterns (PAMPs) and damage-associated molecular patterns (DAMPs) [26,27,28]. In response to the activation signals, seven NLRP3 receptors oligomerize through the NACHT domains and then the apoptosis-associated speck-like protein containing CARD (ASC/PYCARD) adaptor protein interacts with NLRP3 through PYD-PYD interaction that facilitates the formation of the helical ASC filament core. Subsequently, via CARD-CARD interaction, ASC recruits pro-caspase-1 and enables its activation through self-cleavage. Thereafter, active caspase-1 cleaves pro-IL-1β and pro-IL-18 and results in the secretion of their biologically active mature forms [27,28]. The inflammatory cytokines IL-1β and IL-18, generated through NLRP3 activation, play a central role in antibacterial and antifungal inflammatory responses [29,30,31].

Nevertheless, it is also known that NLRP3 polymorphism results in the abnormal activation of NLRP3 inflammasome and the increased secretion of IL-1β and IL-18 by innate immune cells induces systemic inflammation that eventually culminates in chronic tissue damage and in the development of autoinflammatory conditions [32]. Moreover, NLRP3-triggered IL-1β can affect lymphocytes in many different ways; thus, it can serve as an important link for NLR-triggered adaptive immune responses as well. Among others, it promotes B cell proliferation and antibody production and increases T cell survival and polarization, which may also promote the differentiation of autoreactive T helper (Th) 1 and Th17 cells [33]. Therefore, several autoimmune diseases, including SLE, psoriasis, multiple sclerosis (MS), rheumatoid arthritis (RA) or inflammatory bowel diseases (IBD), can be associated with NLRP3 overactivity [33,34,35,36].

Thus, increased activity of both the type I IFN pathway and the IL-1β secretory pathway may lead to autoimmune pathologies in which pDCs may play a key role [37,38,39]. However, it is well known that the two pathways interfere with each other. In particular, type I IFNs inhibit NLRP3 inflammasome-dependent IL-1β production through different mechanisms [40,41,42], which might explain why the body’s antibacterial defense mechanism is weakened and more prone to bacterial superinfections following severe viral infections [43,44,45,46,47]. In addition, IL-1β can also inhibit the type I IFN pathway in multiple ways [48,49,50] that represents a mutually negative interaction between the two pathways. Nevertheless, the interplay between the NLRP3 inflammasome-dependent IL-1β and type I IFN pathway has not been studied in human pDCs yet. Since pDCs are known for their ability to produce massive amounts of type I IFNs [4], we hypothesize that the potential NLRP3-dependent release of IL-1β by pDCs might be initiated during those immune responses when the high IFN-I-producing ability of pDCs is not dominant. In recent years, increasing evidence has indicated that pDCs are more complex than “virus-specific” immune cells. In addition to playing a prominent role in combating viral infections, pDCs are also involved in inflammatory immune responses against various bacteria [51,52,53,54] and fungi [55,56,57,58,59,60]. As mentioned above, these antibacterial and antifungal responses are often coordinated by inflammatory cytokines released through NLRP3 inflammasome activation [61,62,63,64,65].

Therefore, we aimed to explore the baseline and activation-induced expression levels of NLRP3 inflammasome components in human pDCs and we sought to identify those exogenous and endogenous signals, which might induce the assembly of NLRP3 inflammasomes and the subsequent secretion of IL-1 family cytokines in human pDCs. Furthermore, we wanted to explore how the activity of the type I IFN pathway affects NLRP3 inflammasome activation and what type of interaction might exist between the type I IFN and IL-1β pathways in pDCs. Thus, in our work, we investigated the NLRP3 pathway activity in pDCs by using activation signals with high type I IFN-inducing capacity and under pathological conditions with elevated type I IFN signature that might reveal which immunological responses might promote or suppress the NLRP3-mediated inflammatory pathways in pDCs in vivo.

## 2. Results

### 2.1. Investigating the Expression Profile of NLRP3 Pathway Components in a Human pDC Cell Line

In our experiments, we first explored the baseline and activation-induced expression profile of NLRP3 pathway components in the GEN2.2 human pDC cell line at the mRNA level by quantitative polymerase chain reaction (qPCR) (Appendix A) and at the protein level by Western blot (Figure 1A,B). Our results show that the NLRP3 receptor, the ASC adaptor protein and the pro-form of caspase-1 are constitutively expressed both at the mRNA (Appendix A) and the protein levels (Figure 1A,B), whereas the cleaved, active form of caspase-1 is undetectable at the protein level in resting pDCs (data not shown). IL-1β is only marginally expressed at the mRNA level (Appendix A) and while its pro-form is barely detectable at the protein level (Figure 1A,B), its cleaved form is undetectable in unstimulated pDCs (data not shown). Although low levels of IL-18 could be detected at the mRNA level in resting pDCs (Appendix A), its pro-form was not detectable at the protein level (data not shown). IL-1α was not present either at the mRNA or protein level in non-activated pDCs (data not shown).

Next, GEN2.2 cells were activated with various synthetic TLR agonists, which modulated the expression levels of NLRP3 pathway components to different extents. The greatest differences were seen in the effects of the TLR9 ligands, CpG-A and CpG-B. Although these unmethylated oligodeoxynucleotide (ODN) sequences act through the same receptor, CpG-A encounters TLR9 in early endosomes and mostly results in type I IFN secretion, whereas CpG-B is preferentially trafficked to late endosomes and, thus, it is a more potent inducer of NF-κB activation and pro-inflammatory cytokine production [66]. CpG-A exposure significantly decreased NLRP3 expression, both at the mRNA and protein level, whereas CpG-B did not affect its mRNA and protein levels, as compared to unstimulated cells (Appendix A and Figure 1A,B). The level of the NLRP3 adaptor protein ASC was not altered by CpG-A treatment, while a 24 h treatment with CpG-B significantly upregulated its expression at the mRNA level; however, that increase was not detectable at the protein level (Appendix A and Figure 1A,B). Both TLR9 agonists enhanced caspase-1 expression; however, a significant increase was detected only in response to a 24 h exposure to CpG-B, both at the mRNA and protein levels (Appendix A and Figure 1A,B). The major difference between the two ligands was observed in their ability to induce IL-1β production. While CpG-B significantly increased IL-1β production in pDCs at 3 and 6 h, CpG-A was not able to elicit IL-1β expression (Appendix A and Figure 1A,B). Although CpG-B significantly upregulated the expression of IL-18 at the mRNA level, the increase was only about one-tenth of the relative expression of IL-1β and probably, therefore, we were not able to detect IL-18 at the protein level in activated pDCs (Appendix A). Interestingly, the TLR7 agonist imiquimod induced similar changes in the expression of NLRP3 pathway components as CpG-B, with the only difference in IL-1β levels, which was significantly higher not only at 3 and 6 h but also at 12 and 24 h of treatment, as compared to unstimulated cells (Appendix A and Figure 1A,B). These results suggest that different activation signals exhibit distinct IL-1β production kinetics and that pro-IL-1β protein levels are already detectable within 3 h after activation of pDCs.

Multiple lines of evidence indicate that NLRP3 inflammasome-mediated inflammatory responses are of great importance in the control of bacterial and fungal infections. Several bacteria and bacteria-released components, such as various bacterial toxins, provide potent activation signals for inflammasomes [67]. Thus, in our following experiments, pDCs were activated with live pathogenic *Escherichia coli* (*E. coli*), and non-pathogenic bacteria (*Bacillus subtilis*, *Lactobacillus rhamnosus*), and opportunistic fungi (*Candida albicans*). Among the tested microbes, only *E. coli* was able to induce a significant increase in the IL-1β mRNA expression, with peaks at 3 and 6 h that were similar in extent to those observed after CpG-B activation (Appendix A). Furthermore, *E. coli* also provided the strongest activation signal for the secretion of tumor necrosis factor (TNF) and IL-6 pro-inflammatory cytokines and the IL-8 chemokine when compared to the other microbial stimuli (Appendix A). Based on these results, we decided to stimulate pDCs with *E. coli* to investigate the mRNA (Appendix A) and protein (Figure 2A,B) level expression of NLRP3 pathway components. Similar to CpG-B and imiquimod, *E. coli* stimulation enhanced pro-IL-1β expression as early as 3 and 6 h following activation, while it increased pro-caspase-1 expression as late as 12 and 24 h following bacterial stimuli. Further, we found that exposure to *E.coli* did not affect NLRP3 and ASC protein levels (Figure 2A,B) and *E. coli* exposure alone did not induce the cleaved form of caspase-1 or IL-1β (data not shown).

All these data imply that in pDCs, mainly Gram-negative, pathogenic bacteria are capable of inducing the expression of the pro-form of IL-1β. However, in sharp contrast to myeloid cells, such as monocytes with high NLRP3 activity, priming signals alone are not sufficient to induce the production of the cleaved, bioactive form of IL-1β in pDCs.

### 2.2. Assessment of Mature IL-1β Secretion in a Human pDC Cell Line

Next, we wanted to know whether inactive pro-IL-1β could be cleaved and converted into its bioactive, mature form in pDCs. As mentioned above, NLRP3 inflammasome activation generally requires two signals, a primary “priming” signal and a secondary activation signal [26]. The secondary signal ensures the assembly and activation of the NLRP3 inflammasome, which induces the self-cleavage and, thus, the activation of pro-caspase-1 that eventually initiates the cleavage of pro-IL-1β [68]. Numerous secondary signals, including various DAMPs and PAMPs, are known to induce the assembly of the NLRP3 inflammasome [27]. Since we observed that TLR activation alone was not sufficient enough to generate the cleaved form of IL-1β in human pDCs, we applied various secondary activation signals, including adenosine triphosphate (ATP), uric acid sodium salt (MSU), aluminum hydroxide (Alum), the mitochondrial (reactive oxygen species) ROS generator antimycin A [69] and nigericin after priming with CpG-B. Thereafter, we measured the concentration of secreted IL-1β in the cell supernatants in the presence or absence of fetal bovine serum (FBS) in a time-dependent manner (Appendix A). We investigated IL-1β production in the absence of FBS to exclude the ATPase activity of the serum (Appendix A). ATP is rapidly degraded by the transmembrane ecto-ATPases. First, CD39 converts ATP to AMP, which is then dephosphorylated to adenosine by CD73 [70,71]. In addition, the soluble active form of both ectonucleotidases can also be found in the serum [71,72,73,74]; thus, the hydrolysis of nucleotides can take place in the serum as well [75]. Interestingly, among the tested NLRP3 activators, only nigericin, a bacterial toxin, which acts as a potassium ionophore [76], was found to be a potent secondary signal leading to NLRP3 activation in pDCs (Appendix A). In macrophages, ATP is one of the strongest NLRP3 activators; however, our results demonstrated that ATP is not able to induce IL-1β secretion in pDCs. This might be explained by the lack of the ATP-recognizing receptor P2X purinoceptor 7 (P2X7), which is not even inducible by TLR stimulation in pDCs (Appendix A). Based on these results, we used nigericin as a secondary signal in our further experiments. Similar to priming with CpG-B, pretreatment with imiquimod and *E. coli* resulted in significant IL-1β secretion in combination with nigericin. Nevertheless, activation signals, including CpG-A (Figure 3A), *Bacillus subtilis*, *Lactobacillus rhamnosus* and *Candida albicans* (Figure 3B), which were not able to induce the pro-form of IL-1β, did not induce the cleaved form of IL-1β, even in the presence of nigericin. Similar to priming alone (Figure 3A,B, Appendix A), secondary signals alone, including nigericin, were not sufficient to produce cleaved IL-1β in pDCs (data not shown).

In parallel experiments, we utilized the specific NLRP3 inhibitor MCC950 to clarify whether IL-1β secretion by pDCs is indeed NLRP3 inflammasome dependent. As the specific NLRP3 inhibitor almost completely blocked IL-1β secretion by pDCs (Figure 3A,B), we can conclude that mature IL-1β production by pDCs is mainly NLRP3 dependent. In addition, we found that the caspase-1 specific inhibitor Z-YVAD-FMK also significantly reduced the release of mature IL-1β by pDCs (Appendix A). In line with that, we also detected lower cleaved-caspase-1 levels in the cells when the NLRP3 was activated in the presence of the above mentioned inhibitors (Appendix A).

In summary, our results suggest that only the combination of specific priming and secondary signals can induce NLRP3 inflammasome activity in human pDCs.

### 2.3. Analysis of NLRP3 Pathway Activity in Primary Human pDCs

After mapping the expression profile of NLRP3 pathway components in the GEN2.2 human pDC cell line and identifying activation signals that can activate the NLRP3 pathway in these cells, we were curious to see whether a similar activation profile could be detected in primary human pDCs as well. For our experiments, we first collected peripheral blood samples from healthy blood donors, then isolated mononuclear cells from the blood samples by Ficoll-Plaque Plus gradient centrifugation and assessed the expression of NLRP3 pathway components within the gated pDC cell population through flow cytometry using cell surface and intracellular staining. To define the pDC cell population, we first identified the pDC population based on the side scatter (SSC) parameter and blood dendritic cell antigen (BDCA) 4 positivity that showed a homogeneous BDCA4+ population below an approximate SSC value of 200 (Appendix A). Based on SSC and forward scatter (FSC) parameters, the gated BDCA4-positive population is not uniform; thus, we performed a second gating step, where a uniform cell population within the BDCA4 population was gated below approximately 200 on the SSC axis and in the 400–600 region of FSC (Appendix A). Without a second gating step, the BDCA4-positive population also showed some positivity for Linage marker (CD14, CD19, CD56, CD3, CD16 and CD20) (Appendix A), CD11c (Appendix A), CD2 (Appendix A), CD5 (Appendix A), AXL (Appendix A) and CD33 (Appendix A), which are not pDC-specific markers. Furthermore, without a second gating step, only about 90% of the BDCA4-positive population is positive for CD123 (Appendix A), also indicating some degree of contamination with other cell types. However, with a second gating based on SSC-FSC parameters, the positivity/percentage of non-pDC-specific markers within the BDCA4+ population can be significantly reduced (Appendix A). The highest positivity for the non-pDC specific-markers within the BDCA4+ population is shown by those cells, which are located above approximately 600 on the FSC axis (Appendix A). Thus, we suppose that these cells are likely to be more than just possibly clumped pDCs with higher FSC, i.e., size values. Thus, in our further experiments, we used a second gating based on SSC-FSC parameters in addition to BDCA4 positivity to define the pDC population, on which we further examined the expression of NLRP3 pathway components.

Representative dot plots in Figure 4 show that NLRP3, ASC and pro-caspase-1 are expressed constitutively almost in the entire pDC population (Figure 4A). However, ASC protein levels were above 95% in all the donors tested; NLRP3 and pro-caspase-1 levels represented a broader expression range, with some donors showing lower expression values (Figure 4B). At the same time, cleaved-caspase-1, pro-IL-1β and cleaved-IL-1β levels were barely detectable in unstimulated pDCs (Figure 4A,B).

Next, we investigated whether the cleaved form of caspase-1 and the pro-form and cleaved form of IL-1β can be induced in primary pDCs upon activation (Figure 5). The TLR9 ligands CpG-A and CpG-B and the TLR7 agonist imiquimod used alone did not alter pro-caspase-1 levels (Figure 5A) and were not able to induce the cleaved form of caspase-1 (Figure 5B). However, in combination with nigericin, both CpG-B and imiquimod significantly increased cleaved caspase-1 levels in cells, although it is important to note that their efficacy was highly donor dependent (Figure 5B). As observed in the pDC cell line (Figure 3A), pro-IL-1β was mainly detectable by stimulation with CpG-B or imiquimod. The cleaved form of IL-1β was induced only with CpG-B or imiquimod priming following nigericin treatment. On the contrary, CpG-A was found to be a weak activation signal for the induction of both the pro- and cleaved forms of IL-1β in primary pDCs (Figure 5C,D). Similar to the cleaved caspase-1 levels, we detected donor-dependent variations in the extent of IL-1β induction as well (Figure 5C,D). Due to the specific Golgi-independent secretion profile of IL-1β, the intracellular staining process was performed without the use of a protein transport inhibitor [77,78]. However, based on our preliminary results with primary pDCs, 1 h nigericin treatment after TLR priming proved to be the most effective and showed the highest positivity for the cleaved form of IL-1β within the pDC population (data not shown). Using the same experimental conditions, we also validated the intracellular localization of the cleaved form of IL-1β and caspase-1 in pDCs by fluorescence confocal microscopy as well (Appendix A).

These results suggest that similar activation signals are required for NLRP3 activation in primary pDCs and in the GEN2.2 human pDC cell line.

### 2.4. Analysis of NLRP3 Pathway Activity in the Presence of Type I IFNs in pDCs

We hypothesized that the ability of the two TLR9 agonists (CpG-A and CpG-B) to induce IL-1β release by pDCs to a different extent is due to their distinct type I IFN inducing potential [79]. When using the same concentration of the two ligands, CpG-A resulted in a strong and pro-longed secretion of IFN-α (Appendix A), whereas CpG-B induced only very little IFN-α production in GEN2.2 cells (Appendix A). Although both TLR9 ligands could induce the secretion of the pro-inflammatory cytokines IL-6 and TNF, CpG-B resulted in higher cytokine concentrations compared to CpG-A (Appendix A). These results suggest that the major difference between CpG-A and CpG-B is that they have a very different type I IFN-inducing capacity. Thus, we propose that the strong type I IFN triggering capacity of CpG-A may be responsible for its limited potential to induce IL-1β release by human pDCs. In other cell types, a substantial body of evidence suggests that activation of the type I IFN pathway can inhibit the NLRP3-dependent IL-1β pathway by enhancing the expression of several negative regulatory molecules, which inhibit IL-1β formation at the mRNA level, block secondary signals required for NLRP3 activation or directly interact with and, thus, inhibit the assembly of NLRP3 inflammasome components [40,41,80]. Therefore, we activated pDCs with CpG-B in the presence of IFN-α and found that it was able to inhibit pro-IL-1β production, both at the mRNA (Figure 6A) and protein (Figure 6B) levels after 3 and 6 h of CpG-B stimulation. Furthermore, cleaved IL-1β concentrations were also significantly lower when IFN-α was added to cells prior to CpG-B and nigericin treatment (Figure 6C). Thus, it can be concluded that type I IFNs inhibit NLRP3-dependent IL-1β pathway activity in pDCs. Interestingly, in the presence of IL-1β, CpG-A-induced type I IFN production was also significantly reduced, both at the mRNA and protein levels (Appendix A), suggesting a mutual negative interaction between the type I IFN and IL-1β pathways.

### 2.5. Assessing the NLRP3 Activity in pDCs in the Presence of Viruses with High Type I IFN Inducing Capacity

Next, we wanted to investigate the extent to which viruses with strong type I IFN inducing capacity can induce pro-IL-1β production in human pDCs when compared to bacterial stimuli (Appendix A). Thus, GEN2.2 cells were treated with the RNA virus *Vesicular Stomatitis Virus* (VSV), the DNA virus *Herpes Simplex Virus* (HSV) and the pathogenic bacterium *E. coli*, which is a good IL-1β inducer, as our previous results showed. Both RNA and DNA viruses greatly enhanced type I IFN production compared to *E. coli* (Appendix A). Viral exposure did not induce significant pro-IL-1β production, which is in high contrast with *E. coli*, which is a strong inducer of pro-IL-1β after 3 and 6 h of treatment (Appendix A).

We then investigated how *E. coli*-induced IL-1β production of pDCs is affected by the presence of viruses. Therefore, the GEN2.2 pDC cell line was pretreated with VSV or HSV for 3 h, when pDCs already produce considerable levels of IFNs (Appendix A) and then exposed to *E. coli* for 3 or 6 h. Our results showed that pretreatment with viruses significantly reduced *E. coli*-induced IL-1β mRNA expression (Figure 7A) as well as pro-IL-1β protein levels (Figure 7B). Moreover, viral exposure substantially suppressed the release of mature IL-1β by pDCs upon co-treatment with *E.coli* and nigericin (Figure 7C). It is important to highlight, however, that viral pretreatment did not affect the *E. coli*-induced production of TNF, IL-6 and IL-8 (Appendix A).

Subsequently, we repeated our experiments with primary human pDCs and measured IL-1β levels by intracellular staining using flow cytometry. PBMCs from healthy donors were treated with VSV, HSV and *E. coli* for 2.5 h. In the gated primary pDC population, we found that only *E. coli* treatment resulted in significant pro-IL-1β induction, while VSV and HSV did not induce any changes compared to unstimulated cells (Figure 8A). Furthermore, 1 h exposure to nigericin was able to upregulate cleaved IL-1β levels upon *E.coli* pretreatment, whereas it did not induce detectable changes upon prior exposure to viruses (Figure 8B). In addition, neither *E. coli* nor the viruses induced the cleavage of IL-1β in the absence of nigericin. These results suggest that the NLRP3-dependent IL-1β pathway is barely inducible in human pDCs upon viral exposure. 

Furthermore, we found that similar to the pDC cell line, viral pretreatment dampened bacteria-induced IL-1β production in primary pDCs as well. Prior exposure to VSV or HSV viruses significantly suppressed the *E. coli*-induced pro-IL-1β protein levels (Figure 8C) and the *E. coli* and nigericin co-treatment mediated production of cleaved IL-1β (Figure 8D). These results suggest that bacteria-induced IL-1β production by pDCs can be inhibited in the presence of RNA and DNA viruses, probably due to the inhibitory effects of virus-induced type I IFNs on the NLRP3-dependent IL-1β pathway.

### 2.6. Analysis of NLRP3-Dependent IL-1β Pathway Activity in pDCs from Psoriasis Patients Associated with High IFN Signature

Finally, we wanted to explore how IL-1β production by human pDCs is affected under pathological conditions, when cells are exposed to high levels of type I IFNs. To this end, we investigated IL-1β production in pDCs from patients with plaque-type psoriasis, an autoimmune disease associated with excessive type I IFN production and pDC overactivation [81]. Significantly higher levels of IFN-α (Figure 9A) and pro-inflammatory cytokines, including IL-1β, were measured in serum samples from patients with psoriasis compared to healthy donors (Figure 9B). Further, we found that the pDC proportion is significantly lower in PBMCs from psoriatic patients compared with healthy volunteers (Figure 9C) that might be explained by the migration and accumulation of pDCs into the inflamed skin lesions of patients [81]. In addition, we found higher *NLRP3, ASC, caspase-1* and *IL1B* mRNA levels in PBMCs from psoriasis patients compared with healthy individuals (Figure 9D–G). Whereas NLRP3 protein levels were significantly lower in pDCs from psoriatic patients compared to those from healthy controls, we could not detect any differences in the baseline protein levels of ASC and caspase-1 between the two groups (Figure 9H). Cleaved caspase-1, pro-IL-1β and cleaved IL-1β are only marginally expressed by unstimulated cells from both psoriatic and healthy individuals (Figure 9H) and in line with our previous experiments, CpG-A treatment did not affect their expression in pDCs from either psoriatic patients or healthy individuals (Figure 9I). On the contrary, both CpG-B and imiquimod resulted in significantly lower induction of cleaved-caspase-1, pro-IL-1β and cleaved IL-1β in pDCs from psoriatic patients compared to pDCs from healthy donors (Figure 9J,K).

Thus, it is likely that the IL-1β pathway is inhibited in pDCs from patients due to high exposure to type I IFNs. This hypothesis is also supported by data obtained from studying the expression of IFN-inducible regulatory molecules, which inhibit NLRP3-dependent IL-1β secretion. Binding of type I IFNs to their receptors activate the transcription factor signal transducer and activator of transcription (STAT)1, which enhances the expression and activity of several regulatory molecules that can inhibit the NLRP3 pathway at multiple levels [40,41,42]. Such negative regulators include cholesterol-25-hydroxylase (CH25H), IL-10, interleukin-1 receptor antagonist (IL-1RA), inducible nitric oxide synthase (iNOS), suppressor of cytokine signaling (SOCS) 1, PYRIN domain-only proteins (POP) and CARD-only proteins (COP) [40,41,42]. Our results show that activation signals, which induce high type I IFN production, such as CpG-A, highly upregulate the expression of *CH25H*, *SOCS1* and *COP1* in human pDCs. Thus, CpG-A is a potent inducer of NLRP3 inhibitory molecules in contrast to CpG-B (Figure 10A). Furthermore, we found that these suppressor proteins were also expressed at significantly higher levels in PBMCs from psoriasis patients compared to those from healthy controls (Figure 10B). Thus, it is possible that, in the presence of high type I IFN levels, CH25H inhibits IL-1β transcription at the mRNA level, SOCS1 inhibits secondary signals required for NLRP3 activation and COP1 inhibits NLRP3-mediated activation of caspase-1 in psoriatic patients [82,83,84] that, altogether, might be responsible for the dampened IL-1β production by patients’ pDCs.

Finally, we also compared the expression of NLRP3 pathway components in the CD14+ monocyte population from psoriatic and healthy individuals, since inflammatory cytokines produced in excess by circulating monocytes also play an important pathological role in psoriasis [85,86,87]. The ability of monocytes to produce type I IFNs is much lower than that of pDCs [88,89,90,91], yet we still wondered whether NLRP3 activity is also hindered by type I IFNs in the monocyte fraction of psoriatic patients. In the steady state, both monocytes and pDCs express NLRP3 and ASC at similar levels (Appendix A) and although we previously found that pDCs show almost 100% positivity for pro-caspase-1 (Figure 4 and Figure 9H), its expression is almost 100-fold higher in monocytes compared to pDCs (Appendix A), indicating a much higher caspase-1 activity in monocytes. Psoriatic patients exhibit a higher percentage of monocytes in peripheral blood compared with healthy individuals (Appendix A). In contrast to pDCs, NLRP3 protein levels were comparable between the monocyte fractions of psoriatic patients and healthy controls (Appendix A). However, monocytes of both psoriatic patients and healthy individuals display high donor-dependent variations in the protein levels of ASC. We detected significantly elevated ASC expression in the monocytes of psoriasis patients compared with healthy controls (Appendix A). The major difference between the pDC and monocyte populations was that high levels of cleaved caspase-1 can be detected in monocytes from psoriatic patients, even without an extra activation signal, while both the pro- and cleaved forms of IL-1β were at marginal levels (Appendix A). While activation (CpG-A, CpG-B, imiquimod and nigericin) significantly decreased the levels of caspase-1, pro-IL-1β and cleaved IL-1β in pDCs of psoriatic patients compared with healthy controls (Figure 9H–K), it did not affect the monocyte fractions of the two groups (Appendix A). Moreover, in contrast to pDCs, monocytes displayed increased expression of caspase-1, pro-IL-1β and cleaved IL-1β upon CpG-A exposure as well (Appendix A) that might be explained by the poor ability of monocytes to induce type I IFN production in response to these activation signals. All these results suggest that either the low baseline caspase-1 levels or the excessive type I IFN production, which inhibits NLRP3 activity, might be responsible for the reduced NLRP3 pathway activity and IL-1β production in human pDCs.

## 3. Discussion

Due to their unique capacity to rapidly produce massive amounts of type I IFNs, human pDCs are considered as the master regulators of antiviral responses [4,92] and are also involved in the pathogenesis and pathomechanism of various human diseases. Deficiency in type I IFN production by pDCs might lead to persistent viral infections [93,94] or tumor growth due to the lack of IFN-induced anti-tumor activity [95,96], whereas pDC overactivation can result in the development of autoimmune pathologies associated with a high IFN signature [97,98]. Thus, pDC research has primarily focused on PRRs, which provoke type I IFN responses by pDCs, while few data are available regarding the role of the cytosolic inflammasome-forming NLRP3 receptors and the effector functions of NLRP3-triggered inflammatory cytokines, such as IL-1β and IL-18, in pDC-mediated immune responses. 

In 2009, Crozat and colleagues published that both mouse and human pDCs have low antibacterial activity, as they found that NLRs involved in antibacterial innate immunity, including NLRP3, are expressed at low mRNA levels by pDCs compared to BDCA1+ DCs [99]. Among the human blood DC subtypes, caspase-1 mRNA expression was high in BDCA1+ DCs, whereas it was barely detectable in human pDCs and BDCA3+ DCs [99]. Later, this observation was validated at the protein level by Worah and colleagues [100], who found that neither resting pDCs nor R848-activated pDCs nor BDCA3 + DCs had detectable caspase-1 protein levels compared to BDCA1+ DCs (also known as CD1c + DCs and recently referred to as cDC2), which showed high caspase-1 expression, both at baseline and after activation [100]. Another study from last year also identified BDCA1 + DCs as potent IL-1β producers in contrast to pDCs and BDCA3 + DCs (also known as cDC1) [101]. All these studies assume that pDCs, due to their low caspase-1 levels, do not possess inflammasome activity. However, it is important to highlight that these studies compared the pro-caspase-1 protein levels [100] and IL-1β secretion [101] of pDCs to BDCA1 + DCs; thus, the weaker signals of pDCs might be covered by the strong signals of BDCA1 + DCs. 

Nevertheless, a great body of evidence indicates that pDCs are not only “virus-specific” immune cells, as they may play a role in immune responses against bacteria and fungi as well. In particular, pDCs exposed to various bacteria undergo a maturation process and can induce antigen-specific responses as well [51,52,53]. Interestingly, much more is known about the immune response of pDCs against fungi and the mechanism of their antifungal activity is also widely explored [55,56,57,58,59,60]. These data demonstrate that pDC functions are much more diverse than previously thought and that they may be involved in immune responses against pathogens other than viruses as well. However, to date, there are no data pertaining to the role of the NLRP3-dependent IL-1β pathway in the antibacterial or antifungal activities of pDCs.

Therefore, we first explored whether NLRP3 activity can be detected in the human GEN2.2 pDC cell line, which phenotypically and functionally exhibits the characteristics of primary human pDCs [102]. In this cell line, we were able to detect the NLRP3 receptor, the ASC adaptor protein and the pro-caspase-1 enzyme, both at the mRNA and protein levels in resting and activated states. However, among the inflammatory cytokines associated with NLRP3 activity, only the pro-form of IL-1β was detectable both at the mRNA and protein levels in activated cells. IL-18 expression was very low in both resting and activated cells at the mRNA level and was not detectable at the protein level. Although some studies suggest that IL-18 production by mouse pDCs may contribute to NK cell activation and antigen-specific T cell responses upon viral infection, these studies, like ours, failed to detect IL-18 at the protein level in pDCs [103,104]. Furthermore, although the precursor of IL-1α is constitutively present in almost all cell types under physiological conditions [105], we could not detect IL-1α levels in pDCs under any of the tested conditions. Based on literature data, it is hypothesized that a specific cytokine environment and a combination of stimulation signals are required to induce IL-1α production in pDCs, since IL-1α production has so far only been described in tumor-associated pDCs, which are exposed to the effects of tumor-derived immunosuppressive environment [21]. Furthermore, the IL-1α expression was found to depend on AIM2 and not NLRP3 inflammasome activity [21]. These findings suggest that IL-1β may play the major role in the NLRP3-mediated antimicrobial immune responses of human pDCs. 

It is also important to emphasize that we found that distinct activation signals induce different levels of pro-IL-1β expression in pDCs. For example, CpG-B, a ligand for the endosomal TLR9 effectively induces pro-IL-1β, whereas CpG-A, another TLR9 agonist, is not able to do so. This observation is in agreement with literature data showing that different TLR agonists may have distinct IL-1β induction profiles [106]. Furthermore, our results suggest that pro-IL-1β is induced early in pDCs following activation, similar to monocytes and macrophages, which are the main sources of bioactive IL-1β in the myeloid cell compartment [107,108,109,110]. However, the distinct TLR activation signals differed not only in their pro-IL-1β induction ability but also in their pro-IL-1β production kinetics. While pro-IL-1β production reached a peak at 3–6 h after CpG-B stimulation, the TLR7 agonist imiquimod showed prolonged pro-IL-1β production kinetics. Imiquimod has previously been described to mediate complex activation signals in cells and to have TLR7-independent effects as well [111,112,113]. It induces ROS production via the quinone oxidoreductase (NQO) 2 and mitochondrial complex I that may result in K+ efflux-independent NLRP3 activation through the NIMA-related kinase (NEK) 7 adaptor protein [106]. In addition to inducing NLRP3 assembly as a secondary signal, ROS also function as a priming signal and enhance NLRP3 and pro-IL-1β expression by increasing the activity of the NF-κB signaling pathway [114,115]. Thus, it is likely that the ROS-inducing effect of imiquimod may also contribute to the prolonged pro-IL-1β production kinetics in pDCs. Of note, macrophages differentiated from monocytes using various cytokines also exhibit different IL-1β production kinetics [107], suggesting that the type of activation signals has a strong influence on the kinetic profile of IL-1β.

The TLR agonists CpG-B and imiquimod rapidly induced pro-IL-1β production, whereas enhanced pro-caspase-1 protein levels at later incubation time points, mainly at 24 h, when pro-IL-1β induction has already subsided in pDCs. A similar upregulation of pro-caspase-1 levels was observed in CD1c+ DCs after overnight stimulation with R848 [100], whereas in monocytes and macrophages, constitutively high pro-caspase-1 levels were detected independently of the activation signals [107,116]. These data suggest that re-exposing pDCs to TLR activation after 24 h may result in higher NLRP3 inflammasome activity due to significantly higher caspase-1 levels. However, we were not able to test our hypothesis due to the very short lifespan of human pDCs under in vitro conditions.

In general, two activation signals are required for classical, canonical NLRP3 activation [68]. First, the priming signal enhances NLRP3 and pro-IL-1β expression via the NF-κB pathway [23] as well as through the post-translational modification of NLRP3 [117,118] and primes the assembly of inflammasomes, which is then effectively activated by a secondary signal, including whole pathogens, PAMPs or DAMPs [27]. However, an alternative form of NLRP3 activation can be observed in some cell types, such as human monocytes, in which the priming signal alone may also result in NLRP3-dependent mature IL-1β production [119,120]. Furthermore, it has also been described that appropriate secondary signals alone can also evoke classical NLRP3 activation in monocytes [20]. The non-canonical inflammasome promotes pyroptosis and IL-1α release via triggering caspase-4/5 (caspase-11 in mice) activation in a caspase-1 independent manner [41]. Our results showed that in human pDCs, treatment with synthetic TLR ligands alone is insufficient to induce actual NLRP3 activation and mature IL-1β release. It is widely accepted that the TLR4 agonist lipopolysaccharide (LPS), a cell-wall component in Gram-negative bacteria, alone is able to activate NLRP3 inflammasome in monocytes, independently of a second signal [119]. Non-canonical inflammasome activation can also be triggered by cytosolic LPS that induces pyroptotic cell death upon binding to caspase-4/5/11 [41]. In our experiments, we found that the Gram-negative bacteria *E. coli* provides the strongest priming signal for human pDCs when compared to Gram-positive bacteria strains and the opportunistic fungi *Candida albicans*. However, similar to treatments with synthetic TLR ligands, *E. coli* alone did not induce mature IL-1β release, which is likely due to the fact that human pDCs, unlike mouse pDCs, do not express TLR4 [5,121].

Thus, we investigated which secondary signals might induce NLRP3 assembly in the human GEN2.2 pDC cell line following priming and found, that, of the secondary signals we tested, only the potassium ionophore and bacterial toxin, nigericin [76], was able to induce detectable levels of mature IL-1β already after 1 h of treatment. This may suggest that canonical NLRP3 inflammasome activation in pDCs requires potassium efflux. Interestingly, we found that, pDCs do not express the ATP sensing P2X7 receptor, which, in contrast, is highly expressed by monocytes, macrophages and monocyte-derived DCs; thus, it is likely that this potent endogenous danger signal does not function as an NLRP3 activator in pDCs [107,122,123].

After we determined the stimulatory conditions for NLRP3 activation in the human pDC cell line, we repeated our experiments with primary pDCs as well. Recently, it has become apparent that peripheral blood pre-DCs, which can serve as potential precursors for both cDC1 and cDC2 blood DCs, also expresses pDC-specific markers, including CD123, BDCA2 (CD303) and BDCA4 (CD304) proteins, that are commonly used to isolate pDCs [124]. Furthermore, additional markers, such as CD2 [125], CD5 [126], AXL [127] and CD33 [128] proteins, are currently being used to monitor the composition or the purity of the pDC population or even to sort pDCs. Thus, magnetic cell sorting based on BDCA4 expression requires an additional sorting step using a combination of various cell surface markers [54,129,130]. Thus, lately, pDCs have been generally sorted as Lin-CD11c-CD4+CD2-CD5-AXL- cells [131]. In the present study, to avoid cellular stress caused by sorting, which could potentially affect the redox status, metabolic state and viability of cells [132], we identified the pDC population within PBMCs and assessed the activity of NLRP3 pathway components by intracellular staining using flow cytometry. Previously, our laboratory has described that without using any other cell surface markers, BDCA4 positivity combined with light scatter parameters alone is suitable to identify pDCs in peripheral blood [133]. In the present study, we also showed that a second gating based on light scatter parameters applied after gating for BDCA4 can achieve a highly clear pDC population and is adequate to exclude the CD33+ pre-DC population, too. 

In the gated primary pDC population, we observed a similar NLRP3 activation pattern, as previously seen in GEN2.2 cells, i.e., CpG-B and imiquimod were shown to be the strongest priming signals for pro-IL-1β induction and, in combination with nigericin, for mature IL-1β production, compared to CpG-A. Similarly, priming or a secondary signal alone could not induce detectable levels of bioactive IL-1β in primary pDCs. In contrast to our results, some literature data demonstrate that primary pDCs produce detectable levels of IL-1β, even in the absence of secondary signals. One study showed that primary pDCs secrete low levels of IL-1β (approximately 20–35 pg/mL) after a 24 h imiquimod (R837) treatment, [134] and another study found that upon co-culture with epithelial cells, pDCs can serve as a source of IL-1β [135]. Furthermore, the latter study showed IL-1β concentrations as high as 100–200 pg/mL in the supernatants of untreated pDC that would suggest a constantly active IL-1β secretion pathway in resting pDCs [135]; however, that is hard to imagine in the absence of activation-induced pro-IL-1β. Of note, in both papers, a one-step magnetic cell separation based on BDCA4 positivity was used to isolate primary pDCs without any additional sorting steps [134,135], which implies inadvertent contamination with pre-DCs based on recent DC classification models [124]. This contamination may cause false-positive results, especially when experiments target functions majorly associated with myeloid DC activity, such as IL-1β or IL-12 production, and not the unique type I IFN producing capacity of pDCs [136]. For instance, IL-12 production detected in human pDC cultures is a result of contamination with pre-DCs [124]. However, there are currently no data about the IL-1β releasing activity of the pre-DC population. In addition, a recent study also used intracellular labeling and flow cytometry to detect IL-1β in PBMC and demonstrated IL-1β positivity in a gated pDC population after 4 h of CL-087 (TLR7 ligand) treatment without using secondary activation [88]. However, it should be pointed out that the anti-IL-1β antibody used by the authors can detect both the pro- and mature form of the cytokine, according to the manufacturer’s protocol; thus, it is likely that the pro-form caused the positivity for IL-1β [88]. Furthermore, the CD123 and CD303 (BDCA2) markers used to identify pDCs in PBMC are not enough to exclude the presence of contaminating DC populations [92,124].

Our further results also highlighted that TLR ligands, which induce a high type I IFN response, such as CpG-A, are unable to induce IL-1β production in pDCs, in contrast to CpG-B. Although both CpG-A and CpG-B activate TLR9, the two ODN sequences elicit different responses by pDCs due to their structural differences. CpG-A ODNs are able to form higher-order multimeric structures due to their palindromic sequence and poly(G) motifs, whereas CpG-B ODNs form linear structures and lack the potential to form multimers [79]. Therefore, CpG-A molecules are retained in early endosomes and lead to high levels of type I IFN secretion in pDCs via activation of the transcription factor interferon regulatory factor (IRF) 7. On the contrary, monomeric CpG-B is rapidly transported to late endosomes, where it mainly facilitates pro-inflammatory cytokine production via NF-κB activation and exhibits a very low potential to induce type I IFN production in pDCs. Therefore, we hypothesized that the strong ability of pDCs to produce type I IFN may also contribute to the low NLRP3 inflammasome activity, as compared to myeloid cells. Indeed, we found that, when pDCs were pretreated with recombinant human IFN-α or viruses inducing type I IFNs, the potent NLRP3 activators of pDCs, including CpG-B or *E. coli*, were unable to induce large amounts of IL-1β production. Studies using other cell types also indicate that type I IFNs can inhibit NLRP3 activation [40,41,80]. It is also known that bacterial superinfections are common after viral infections that may be partly due to the inhibitory effect of the virus-triggered IFN pathway on the bacteria-induced IL-1β pathway [41]. Interestingly, vice versa, IL-1β can also inhibit the secretion of type I IFNs by enhancing the production of eicosanoid lipid mediators, such as prostaglandin E2 [48]. In line with that, we observed that the presence of IL-1β inhibits CpG-A-mediated IFN-α production in pDCs. All these data indicate that viral superinfections can also develop after severe bacterial infections. It is interesting to note, for example, that IFN-α/β therapy for viral hepatitis was ineffective in 60% of cases that could be due to the IL-1β-mediated inflammation in these patients. It was demonstrated using a hepatic cell model system that the antagonistic effect of IL-1β on the antiviral activity of IFN-α/β is owing to their potential to inhibit IFN-α/β-activated STAT1 and downregulate the expression of antiviral proteins [49]. It has also been observed that ASC and caspase-1-deficient DCs and macrophages have increased levels of type I IFNs [137] and NLRP3 or caspase-1-deficient mice also exhibited a greater type I IFN response [138].

Interactions between type I IFN and IL-1β pathways may play an important role not only during pathogen-induced inflammation, but also in the regulation of the pathomechanisms of various autoimmune diseases. Furthermore, the inhibitory effect of type I IFNs on NLRP3 inflammasome activity may also be an important therapeutic option in the treatment of certain pathologies. For example, in patients with MS with increased NLRP3 activity, IFN-β therapy could suppress the increased activity of the inflammasome that might help to reduce the patients’ symptoms [83]. Both pDCs and NLRP3 receptor overexpression play a role in the pathogenesis and pathomechanism of various autoimmune diseases. Numerous single nucleotide polymorphisms (SNPs) of NLRP3, which are mainly gain-of-function mutations, have been described, for example, in SLE, IBD, MS, RA and psoriasis [139,140]. In addition, pDCs may also play a role in the pathogenesis of psoriasis, since self-nucleic acids released by dying cells in complex with overproduced LL37 cationic antimicrobial peptides induce the persistent activation and excessive type I IFN production of pDCs, which promotes the development of T cell-mediated autoimmune responses [141,142]. Therefore, psoriasis is characterized by a high IFN signature [81]. In the present work, we showed that pDCs from psoriasis patients have significantly lower activation-induced NLRP3 activity compared to healthy subjects, underlying our hypothesis that pDCs with high type I IFN activity have low NLRP3 activity due to the inhibitory effects of type I IFNs on the NLRP3 pathway.

By binding to their receptors, type I IFNs can activate the STAT1 transcription factor, which can increase the expression of several negative regulatory molecules, including CH25H, IL-10, IL1RA, iNOS, SOCS1, POP or COP molecules, which either inhibit IL-1β formation at the mRNA level or suppress secondary signals required for NLRP3 activation or directly interact with NLRP3 inflammasome components and inhibit their assembly [40,41,80]. Of these inhibitory mechanisms, IFN-induced IL-10 inhibition is unlikely to occur in pDCs, as data suggest that pDCs are unable to produce IL-10 [143]. Furthermore, pDCs are also not characterized by nitric oxide (NO) synthesis, as they have very low levels of iNOS and, thus, it is unlikely that NO would inhibit NLRP3 through nitrosylation [144]. However, we found that in human pDCs, the expression of *CH25H*, *SOCS1* and *COP1* is significantly increased in the presence of high IFN response-inducing activation signals and we found that the levels of these inhibitory molecules are significantly higher in pDCs from psoriasis patients with high type I IFN levels compared with those from healthy controls. Therefore, we suppose that CH25H inhibits IL-1β transcription, SOCS1 blocks NLRP3 activation and COP1 suppresses NLRP3-dependent activation of caspase-1 in pDCs with high type I IFN pathway activity [82,83,84].

In addition, we found that the overall expression of NLRP3, ASC, IL-1β and caspase-1 is higher in the peripheral blood of psoriatic patients compared to normal controls, implying that overactivation of the NLRP3 inflammasome may also contribute to psoriasis-associated inflammatory responses [85,145,146,147]. In addition, examining the CD14+ monocyte population of patients, we also observed elevated baseline cleaved caspase-1 expression, as compared to healthy controls. It is well known that monocytes have a higher and presumably constant caspase-1 activity compared with macrophages [116,148]. When we compared the levels of NLRP3 pathway components in pDCs and monocytes, we found that the frequency of NLRP3, ASC and pro-caspase-1-positive cells was similar between the two cell populations. Interestingly, the level of pro-caspase-1 expression is significantly higher in monocytes compared to pDCs that supports the idea that the caspase-1-dependent IL-1β production capacity of pDCs may indeed be much lower compared to myeloid cells. However, there is no difference in the expression values of NLRP3 and ASC between the two cell types.

Furthermore, in contrast to pDCs, monocytes from psoriasis patients do not show a reduced NLRP3 activity in response to the same activation signals. The reason behind this might be that the NLRP3 activity of monocytes is not affected as strongly as of pDCs by the inhibitory effects of type I IFNs, since their capacity to produce type I IFNs is several orders of magnitude lower compared to pDCs [1,149]. This is also supported by our findings demonstrating that in contrast to pDCs, CpG-A was able to induce pro-IL-1β, cleaved IL-1β and cleaved caspase-1 expression in monocytes. Several studies have already shown that TLR activation does not or only very slightly induces type I IFN release by monocytes [88,89,90,91]. Although literature data suggest that activated monocytes from psoriasis patients show significantly higher NLRP3 activity than healthy controls and that also contributes to the disease pathology [85,86,87], we could not detect differences in NLRP3 activity between the two groups. It is likely that in our present experiments, we were unable to detect increased NLRP3 activity in patients’ monocytes because we used activation signals tailored to pDCs and, though monocytes express TLR7 [150] and TLR9 [151,152,153] receptors, these cells cannot be strongly stimulated through them. Furthermore, the secretion [154,155] and the constant activity [116,148] of caspase-1 in monocytes also complicate intracellular protein analysis by flow cytometry.

In conclusion, we revealed that human pDCs also express NLRP3 pathway components, the expression of which is differentially affected by distinct activation signals. Furthermore, we show that NLRP3 is functional in these cells, as mature IL-1β secretion can be induced in human pDCs by using a combination of priming and specific secondary activation signals. Nevertheless, we also demonstrated that NLRP3 activity in pDCs is much lower than that of other myeloid cells, due to their low baseline caspase-1 expression and high type I IFN pathway activity. In fact, activation stimuli that induce a high type I IFN response in pDCs also increase the expression of various regulatory molecules, inhibiting NLRP3-mediated IL-1β secretion. Thus, the IL-1β-mediated immune response of pDCs may predominantly prevail under conditions that are not associated with increased type I IFN pathway activity. Our results also indicate that the interplay between the type I IFN and IL-1β pathways is an important regulator of both pathogen-induced and sterile autoimmune inflammatory reactions.

## 4. Materials and Methods

### 4.1. GEN2.2 Cell Line

The human pDC cell line (GEN2.2) [102] used in our experiments was provided by Dr. Joel Plumas and Dr. Laurence Chaperot (Research and Development Laboratory, French Blood Bank Rhône-Alpes, Grenoble, France) and was deposited with the CNCM (French National Collection of Microorganism Cultures) under the number CNCMI-2938. GEN2.2 cells were grown on a layer of mitomycin C (Sigma-Aldrich, St. Louis, MO, USA, Cat. No. M4287)-treated murine MS5 feeder cells (Leibniz Institute DSMZ-German Collection of Microorganisms and Cell Cultures, Braunschweig, Germany, Cat. No. ACC 441) in RPMI 1640 medium (Sigma-Aldrich, Cat. No. R8758) supplemented with 10% heat-inactivated FBS (Life Technologies Corporation, Carlsbad, CA, USA, Cat. No. 10270-106), 100 U/mL penicillin, 100 μg/mL streptomycin (both from Biosera, Nuaille, France, Cat. No. XC-A4122/100) and 5% non-essential amino acids (Life Technologies Corporation, Cat. No. 11140050). For experiments, the GEN2.2 cells were removed from the feeder layer and seeded on 24-well plates at a concentration of 5 × 10^5^ cells/500 μL in complete RPMI 1640 medium (Sigma-Aldrich). Cell lines were grown and incubated at 37 °C in a 5% CO_2_ humidified atmosphere.

### 4.2. Collection of Human Blood Samples and Isolation of PBMCs

Patient blood samples were provided to us by the Laboratory of Regenerative Medicine and Cellular Pharmacology (Department of Dermatology and Allergology, University of Szeged, Szeged, Hungary). The collection of peripheral blood samples complied with the guidelines of the Helsinki Declaration and was approved by the National Public Health and Medical Officer Service (NPHMOS) and the National Medical Research Council (administrative number: 13740-5/2021/EÜIG and 4969; 90/2021-SZTE IKEB, protocol code: PSO-CELL-01) which follows the EU Member States’ Directive 2004/23/EC and GDPR on presumed written consent practice for tissue collection.

Men or women aged 25 to 65 years suffering from plaque psoriasis with a psoriasis area sensitivity index (PASI) greater than 15 were included in the study. Patients who received systemic treatment (biological or traditional) or total body phototherapy were excluded. Peripheral blood of healthy, gender and age-matched donors was used as a control in experiments with psoriasis samples. To characterize the NLRP3 pathway in healthy individuals, 23–55-year-old volunteers were selected for peripheral blood donation.

Then, 25 mL of peripheral blood was collected from psoriatic and healthy donors into 10 mL BD Vacutainer^®^ Plastic whole blood tubes with spray-coated K2EDTA (BD Biosciences, San Jose, CA, USA, Cat. No. 366643). The blood was diluted 1:1 with physiological saline (NaCl 0.9%, B. Braun, Melsungen, Germany) and then PBMCs were separated by Ficoll-Paque Plus (Cytiva Sweden AB, Uppsala, Sweden, Cat. No. 17144003) gradient centrifugation. Freshly isolated PBMCs were seeded in FACS tubes at a density of 1 × 10^6^ cells/ 500 mL RPMI 1640 medium (Sigma-Aldrich) supplemented with 10% heat-inactivated FBS (Life Technologies Corporation), 2 mM L-glutamine, 100 U/mL penicillin and 100 μg/mL streptomycin (all from Biosera).

### 4.3. Stimulation and Treatments of the Cells

To prime the NLRP3, inflammasome cells were incubated with 1 μM CpG-A (ODN 2216; Hycult Biotech, Uden, The Netherlands, Cat. No. HC4037), 1 μM CpG-B (ODN 2006; Hycult Biotech, Cat. No. HC4039) or 5 μg/mL imiquimod (IMQ) (InvivoGen, San Diego, CA, USA, Cat. No. tlrl-imq) in fresh, complete RPMI 1640 medium for the indicated times. In order to activate the inflammasome secondary activation signals including 20 μM nigericin (InvivoGen, Cat. No. tlrl-nig), 5 mM ATP (Invivogen, Cat. No. tlrl-atpl), 250 µg/mL Alum (Thermo Scientific, Cat. No. 77161), 300 µg/mL MSU (Sigma-Aldrich, Cat. No. 1198-77-2) or antimycin A (0.5 µg/mL, Sigma-Aldrich, Cat. No. A8674) were added to the cells after priming without a washing step and then the cells were incubated for 1, 2, 4 or 6 h. 

In separate experiments, cells were primed with *Escherichia coli* (*E. coli*) ATCC11775, *Bacillus subtilis* ATCC6051, *Lactobacillus rhamnosus* ATCC53103, *Candida albicans* ATCC10231, *Vesicular Stomatitis Virus* (VSV; Indiana strain [156]) or *Herpes Simplex Virus 1* (HSV; KOS serotype, ATCC-VR-1493) at a MOI of 10 for bacteria and viruses or 0.01 for *Candida* for the indicated times. Bacteria and *Candida* were provided by Dr. Walter Pfliegler (Department of Molecular Biotechnology and Microbiology, Faculty of Science and Technology, University of Debrecen, Debrecen, Hungary). Viruses were kindly provided by Dr. Eszter Csoma (Department of Medical Microbiology, Faculty of Medicine, University of Debrecen, Debrecen, Hungary). After priming with microbes, cells were incubated with 20 μM nigericin (InvivoGen, Cat. No. tlrl-nig) for the indicated times. In some experiments cells were pretreated with VSV and HSV for 3 h before *E. coli* and nigericin (InvivoGen) treatments.

In parallel experiments 1 µM MCC950 (Invivogen, Cat. No. inh-mcc) or 20 µM Z-YVAD-FMK (BioVision Incorporated, Milpitas, CA, USA, Cat. No. 1012-100) was added to the cells in the last 30 min of the priming and then cells were exposed to the secondary signals without any washing step.

The experiments performed with CpG-B were repeated in the presence of 50 μg/mL recombinant human IFN-α (Abcam, Cambridge, UK, Cat. No. ab48750) as well. In detail, IFN-α was added to the cells 30 min before the CpG-B priming and then the cells were exposed to nigericin without a washing step. Similarly, the experiments with CpG-A were repeated in the presence of 10, 50 or 100 ng/mL recombinant human IL-1β (Peprotech, Cat. No. 200-01B). IL-1β was added to the cells 30 min before the CpG-A treatment.

### 4.4. Generation and Activation of Primary Macrophages

Human heparinized leukocyte-enriched buffy coats were obtained from healthy blood donors drawn at the Regional Blood Center of Hungarian National Blood Transfusion Service (Debrecen, Hungary) in accordance with the written approval of the Director of the National Blood Transfusion Service and the Regional and Institutional Ethics Committee of the University of Debrecen, Faculty of Medicine (Debrecen, Hungary). PBMCs were separated from buffy coats by Ficoll-Paque Plus (Cytiva) gradient centrifugation. Monocytes were purified from PBMCs by positive selection using magnetic cell separation with anti-CD14-conjugated microbeads (Miltenyi Biotech, Bergish Gladbach, Germany, Cat. No. 130-050-201). Monocytes were cultured in 24-well cell culture plates at a density of 5 × 10^5^ cells/500 μL in RPMI 1640 medium (Sigma-Aldrich) supplemented with 10% heat-inactivated FBS (Life Technologies Corporation), 2 mM L-glutamine (Biosera), 100 U/mL penicillin, 100 μg/mL streptomycin (both from Biosera) and 50 ng/mL recombinant human M-CSF (Peprotech, Cat. No. 300-25), then activated with 250 ng/mL Ultrapure LPS, *E. coli* 0111:B4 (Invivogen, Cat. No. tlrl-3pelps) for the indicated times. Cells were incubated at 37 °C in a 5% CO_2_ humidified atmosphere.

### 4.5. Flow Cytometric Analysis

PBMCs were stained with anti-CD14-FITC (BioLegend, San Diego, USA, Cat. No. 301804) and anti-BDCA4-APC (CD304, Neuropilin-1, BioLegend, Cat. No. 354506), then were fixed and permeabilized with BD Cytofix/CytopermTM Plus Fixation/Permeabilization Kit (BD Biosciences, San Diego, CA, USA, Cat. No.554714) according to the manufacturer’s instructions. After washing the cells were incubated with anti-NLRP3 (clone D2P5E, Cat. No. 13158S), anti-ASC (clone E1E3I; Cat. No. 13833S), anti-IL-1β (clone D3U3E; Cat. No. 12703S), anti-cleaved IL-β (Asp116; clone D3A3Z; Cat. No. 83186S), anti-caspase-1 (clone D7F10; Cat. No. 3866S) and anti-cleaved caspase-1 (Asp297; clone D57A2; Cat. No. 4199S) antibodies (all from Cell Signaling, Danvers, MA, USA) for 30 min. After washing the cells were incubated with PE-conjugated IgG1 secondary antibody (Cat. No. 406421; BioLegend) for 30 min. As an isotype control rabbit IgG (Cell Signaling, clone DA1E, Cat. No. 3900S) was used. As a final step, cells were taken up in 2% paraformaldehyde (Alfa Aesar, Haverhill, MA, USA, Cat. No. J61899). Fluorescence intensities were measured with FACSCalibur flow cytometer (BD Biosciences) and data were analyzed with FlowJo software (TreeStar, Ashland, OR, USA). Throughout data acquisition, 600,000 events were acquired from each sample containing 1,000,000 stained cells. Threshold of FSC was set to 150 to exclude cell debris. Monocytes were identified as CD14-positive cells. pDCs were first recognized as being BDCA4 positive. After this step, the distribution of the cells within the BDCA4+ gate was analyzed by back gating on the light scatter parameters to exclude non-pDCs. BDCA4+ cells in the region 400–600 of FSC were identified as pDCs. 

To analyze the phenotype of gated pDCs the following antibodies were used: LIN-FITC (BD Biosciences, Cat. No. 340546), CD2-FITC (Immunotech-Beckman Coulter, Marseille, France, Cat. No. PN IM0442), CD5-PE (PharMingen-BD Biosciences, Cat. No. 555353), CD11c-PE (Immunotech-Beckman Coulter, Cat. No. PN IM1760U), AXL-Alexa Fluor^TM^ 488 (R&D Systems Minneapolis, MN, USA, Cat. No. FAB154G), CD33-FITC (eBioscience, Cat.No. 11-0337-42) and CD123-PE-Cy7 (Biolegend, Cat. No. 301804). 

### 4.6. Confocal Microscopy

Activated PBMCs were stained with anti-BDCA4-APC (CD304, Neuropilin-1, BioLegend, Cat. No. 354506) for 30 min at 4 °C then were fixed and permeabilized with BD Cytofix/CytopermTM Plus Fixation/Permeabilization Kit (BD Biosciences) according to the manufacturer’s instructions. After washing cells were incubated with anti-cleaved IL-1β (Asp116; clone D3A3Z; Cat. No. 83186S) or anti-cleaved caspase-1 (Asp297; clone D57A2; Cat. No. 4199S) antibodies (all from Cell Signaling) for 30 min at 4 °C. After washing cells were incubated with PE-conjugated IgG1 secondary antibody (Cat. No. 406421; BioLegend) for 30 min at 4 °C. As an isotype control rabbit IgG (Cell Signaling, clone DA1E, Cat. No. 3900S) was used. Stained cells were dissolved in Mowiol^®^ 4–88 mounting media (Sigma-Aldrich, Cat. No. 81381).

For confocal microscopy 100,000 cells were pipetted on 8-well μ-Slides (chambered coverslip, ibidi GmbH). Cells were visualized immediately by a Zeiss LSM880 confocal microscope. BDCA4-APC was excited at 633 nm and IgG1-PE was excited at 543 nm. Fluorescence emission was detected through 650 to 670 nm and 560 to 615 nm band-pass filters. Images were taken in multi-track mode to prevent cross-talk. Image stacks of 1024 × 1024 pixel, 1.5 μm thick optical sections were obtained with a 40× C-Apochromat water immersion objective (NA = 1.2). 

### 4.7. Quantitative Real-Time PCR

Total RNA was isolated from 5 × 10^5^ GEN2.2 cells or 1 × 10^6^ PBMCs using Tri Reagent (Molecular Research Center, Inc., Cincinnati, OH, USA, Cat. No. TR118). Total RNA was treated with DNase I (Thermo Fisher Scientific, Waltham, MA, USA, Cat. No. AM2222) to exclude amplification of genomic DNA then reverse transcribed into cDNA using the High-Capacity cDNA RT Kit of Applied Biosystems (Thermo Fisher Scientific, Cat. No. 4374966). Gene expression assays were purchased from Thermo Fisher Scientific for NLRP3 (Assay ID: Hs00918082_m1, Cat. No. 4331182), IL-1β (Assay ID: HS00174097_m1, Cat. No. 4331182), IL-18 (Assay ID: Hs01038788_m1, Cat. No. 4331182), IL-1α (Assay ID: Hs00174092_m1, Cat. No.), caspase-1 (Assay ID: Hs00354836_m1, Cat. No. 4331182), PYCARD/ASC (Assay ID: Hs00203118_m1, Cat. No. 4331182), P2X7 (Assay ID: Hs00175721_m1, Cat. No. 4331182), CH25H (Assay ID: Hs02379634_s1, Cat. No. 4331182), SOCS1 (Assay ID: Hs00705164_s1, Cat. No. 4331182), CARD16 (Assay ID: Hs00430993_m1, Cat. No. 4331182), IL1RN (Assay ID: Hs00893626_m1, Cat. No. 4331182), NOS1 (Assay ID: Hs00167223_m1, Cat. No. 4331182) and Integrated DNA Technologies (Coralville, IA, USA) for IFNA1 (Assay ID: Hs.PT.49a.3184790.g) and PPIA (cyclophilin A; Assay ID: Hs.PT.58v.38887593.g). Quantitative PCR was performed using the ABI StepOne Real-Time PCR System (Thermo Fisher Scientific) and cycle threshold values were determined using the StepOne v2.1 Software (Thermo Fisher Scientific). The relative amount of mRNA was obtained by normalizing to the PPIA housekeeping gene in each experiment.

### 4.8. Western Blotting

For Western blotting 5 × 10^5^ GEN2.2 cells were lysed in Laemmli buffer, heated at 100 °C for 10 min, separated on 10 or 15% SDS-PAGE then electro-transferred to nitrocellulose membranes (Bio-Rad Laboratories GmbH, Munich, Germany). Non-specific binding sites were blocked with 5% non-fat dry milk diluted in TBS Tween buffer (50 mM Tris, 0.5 M NaCl, 0.05% Tween-20, pH 7.4). Membranes were probed with the anti-NLRP3 (clone D2P5E, Cat. No. 13158S), anti-ASC (clone E1E3I; Cat. No. 13833S), anti-IL-1β (clone D3U3E; Cat. No. 12703S) and anti-caspase-1 (clone D7F10; Cat. No. 3866S) (all from Cell Signaling). Beta-actin was used as a loading control (Santa Cruz Biotechnology, clone C4; Cat. No. sc-47778). The bound primary antibodies were conjugated with anti-mouse (Bio-Rad, Cat. No. 1721011) or anti-rabbit (GE Healthcare, Cat. No. NA934) horseradish peroxidase-conjugated secondary antibodies at a dilution of 1:5000 and 1:10,000, respectively, and were visualized by the ECL system using SuperSignal West Pico or Femto chemiluminescent substrate (Thermo Fisher Scientific, Rockford, IL, United States) and X-ray film exposure. Densitometric analysis of immunoreactive bands was performed using Image Studio Lite Software version 5.2 (LI-COR Biosciences, Lincoln, NE, United States).

### 4.9. ELISA

Cell culture supernatants were collected at the indicated time points and the IL-1β (Cat. No.557953), TNF (Cat. No. 555212), IL-6 (Cat. No. 555220) and IL-8 (Cat. No. 555244) levels were determined by the BD OptEIA human ELISA kits (all from BD Biosciences, San Diego, CA, USA). IFN-α levels were measured by the VeriKineTM Human Interferon Alpha ELISA kit (PBL Interferon Sources, Piscataway, NJ, USA). Assays were performed according to the manufacturer’s instructions. Absorbance measurements were carried out by a Synergy HT microplate reader (Bio-Tek Instruments, Winooski, VT, USA) at 450 nm.

### 4.10. Statistical Analysis

Data are expressed as the mean ± SD and analyzed by ANOVA, followed by Bonferroni post hoc test or paired Student *t* test. Data analyses were performed using GraphPad Prism v.6 software (GraphPad Software Inc., La Jolla, CA, USA). Differences were considered to be statistically significant at *p* < 0.05.

## 5. Conclusions

In the current study, we focused on pDCs, which are generally protective immune cells and specialized for the elimination of invading viruses. However, pDCs can play a deleterious role in the pathogenesis of various autoimmune disorders, when the overactivated pDCs respond to self-molecules of the body and facilitate immune responses against healthy tissues, leading to chronic inflammation and extensive tissue damage. The activation of pDCs is primarily mediated by their well-known pattern recognition receptors; however, currently, little is known about the expression of inflammasome-forming receptors in pDCs so far. A growing body of evidence indicates that the excessive activation of inflammasomes can also contribute to the development of various autoimmune diseases. Moreover, there remains controversy in the scientific literature over whether or not inflammasomes and the IL-1β pathway are functional in pDCs. Our present findings demonstrate that the NLPR3 inflammasome can be active in pDCs; however, a specific combination of priming and secondary signals is required for the NLRP3-depedent IL-1β production of pDCs. Nevertheless, compared to the high type I IFN producing capacity of pDCs, their IL-1β secretion is overshadowed by that of other myeloid cell types due to the inhibitory effects of type I IFNs on NLRP3 activity. Therefore, a deeper insight into the regulatory mechanisms of this process would allow us to better understand the features of pDCs under various conditions and especially in pDC-associated disorders.

## Figures and Tables

**Figure 1 ijms-23-12154-f001:**
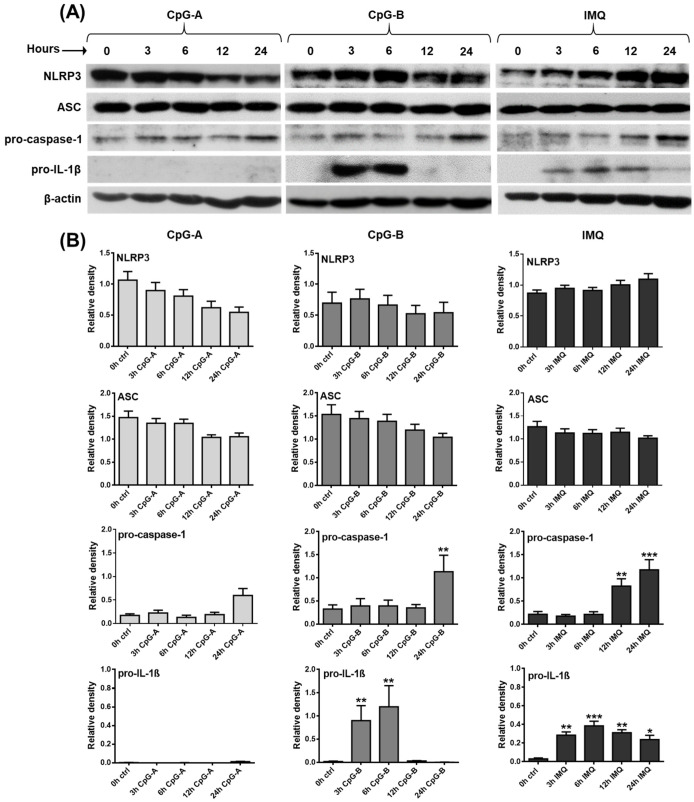
Distinct TLR agonists have different effects on the expression of NLRP3 inflammasome pathway components in the human GEN2.2 pDC cell line. GEN2.2 cells were treated with 1 μM CpG-A, 1 μM CpG-B or 5 μg/mL IMQ in a time-dependent manner (**A**,**B**). The expression of NLRP3, ASC, pro-caspase-1 and pro-IL-1β was measured at the protein level by Western blotting. Representative blots are shown in (**A**). Data are shown as mean ± SD from 5 independent experiments in panel (**B**). Data were analyzed using one-way ANOVA followed by Bonferroni’s post hoc test. * *p* < 0.05, ** *p* < 0.01, *** *p* < 0.001 vs. control (ctrl). IMQ: imiquimod.

**Figure 2 ijms-23-12154-f002:**
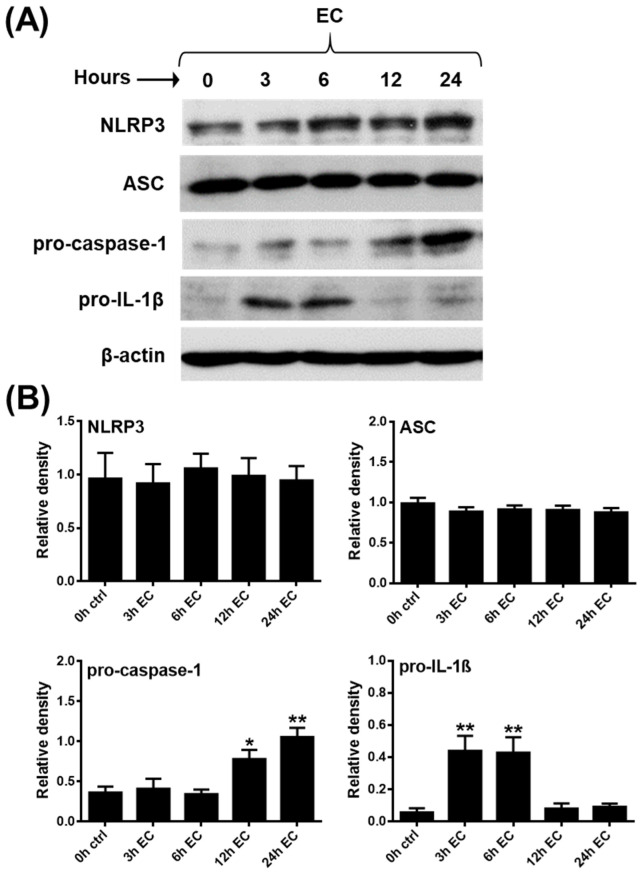
Pathogenic bacteria influence the expression of NLRP3 inflammasome pathway components in the human GEN2.2 pDC cell line. GEN2.2 cells were treated with *E. coli* (MOI 10) in a time-dependent manner (**A**,**B**). The expression of NLRP3, ASC, pro-caspase-1 and pro-IL-1β was measured at the protein level by Western blotting. Representative blots are shown in (**A**). Data are shown as mean ± SD from 5 independent experiments in (**B**). Data were analyzed using one-way ANOVA followed by Bonferroni’s post-hoc test. * *p* < 0.05, ** *p* < 0.01 vs. control (ctrl). EC: *E. coli*.

**Figure 3 ijms-23-12154-f003:**
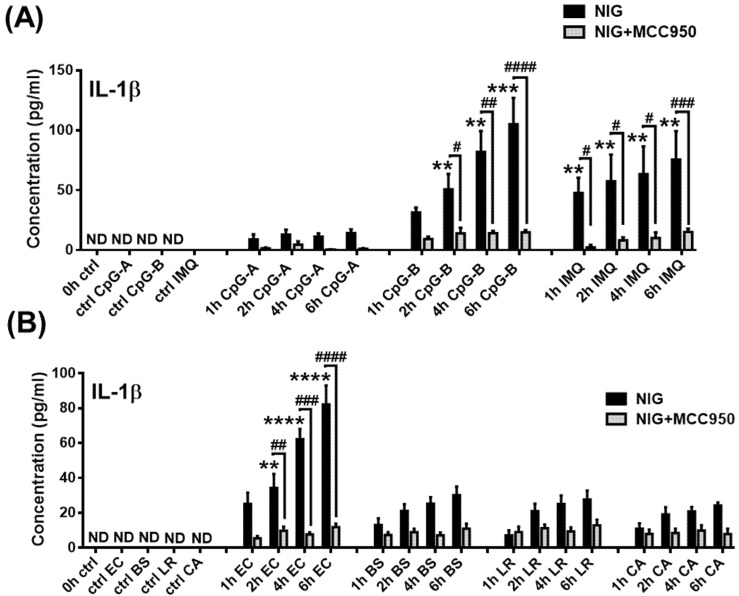
A secondary stimulus is required for the secretion of NLRP3-dependent IL-1β in the human GEN2.2 pDC cell line. GEN2.2 cells were pretreated with 1 μM CpG-A, 1 μM CpG-B or 5 μg/mL IMQ for 3 h, then 20 μM nigericin was added to the cells in a time-dependent manner (**A**). In parallel experiments, cells were incubated in the presence of 1 μM MCC950 (specific NLRP3 inhibitor) for the last 30 min of TLR stimulation prior to nigericin exposure (**A**). In similar experiments, GEN2.2 cells were primed with live *E. coli* (MOI 10), *Bacillus subtilis* (MOI 10), *Lactobacillus rhamnosus* (MOI 10) and *Candida albicans* (MOI 0.01), then nigericin was added to the cells in a time-dependent manner (**B**). Similarly, in a parallel experiment, cells were incubated with MCC950 for the last 30 min of the microbial stimulation prior to nigericin treatment (**B**). IL-1 β secretion was measured by ELISA (**A**,**B**). Data are represented as mean ± SD of 6 individual experiments and were analyzed using one-way ANOVA followed by Bonferroni’s post-hoc test. ** *p* < 0.01, *** *p* < 0.0001, **** *p* < 0.0001 vs. control (ctrl), # *p* < 0.05, ## *p* < 0.01, ### *p* < 0.0001, #### *p* < 0.0001. ND: not determined, IMQ: imiquimod, NIG: nigericin, EC: *Escherichia coli*, BS: *Bacillus subtilis*, LR: *Lactobacillus rhamnosus*, CA: *Candida albicans*.

**Figure 4 ijms-23-12154-f004:**
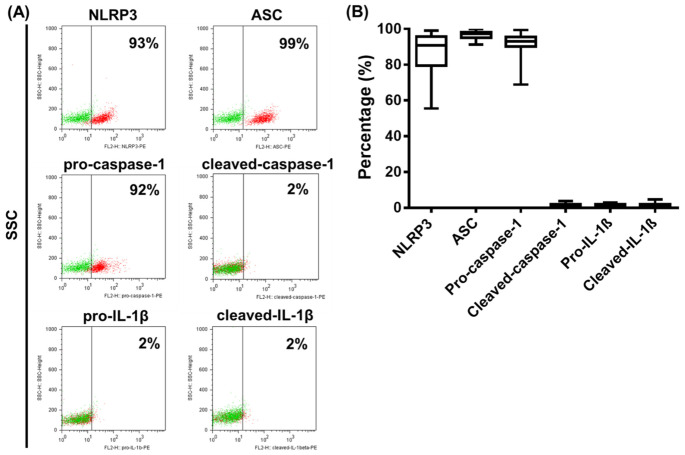
The baseline expression of NLRP3 inflammasome pathway components in primary human pDCs. The NLRP3, ASC, pro-caspase-1, cleaved caspase-1, pro-IL-1β and cleaved IL-1β expressions in primary human pDCs gated from PBMCs were measured using intracellular flow cytometry. Representative dot plots are shown in (**A**). Green dots indicate isotype controls and red dots show the specific proteins (A). Data are represented as mean ± SD of 22 individual experiments and analyzed using one-way ANOVA followed by Bonferroni’s post-hoc test (**B**). SSC: side scatter.

**Figure 5 ijms-23-12154-f005:**
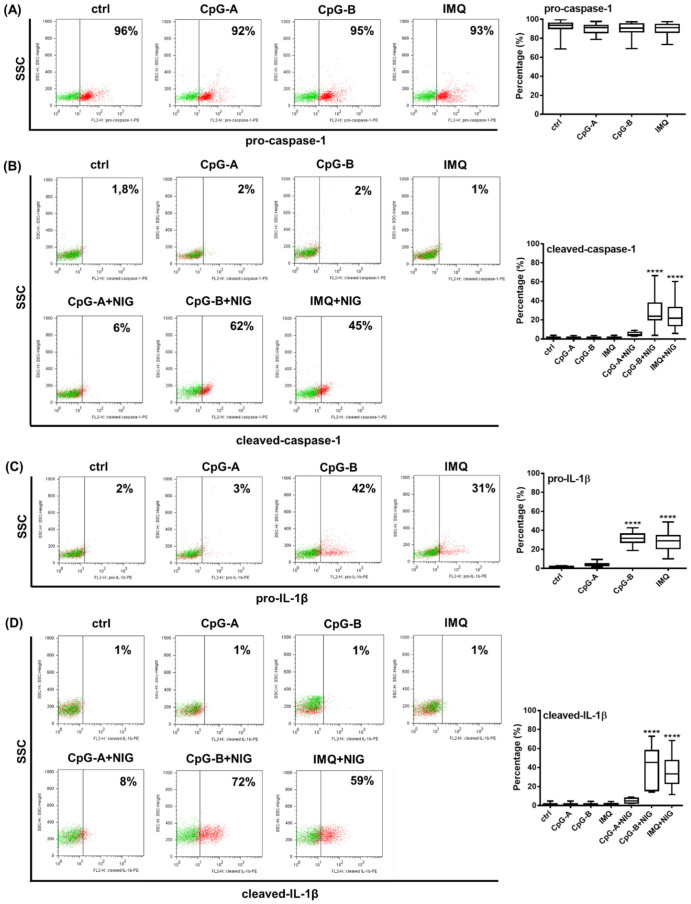
TLR ligands differ in their IL-1β inducing capacity in primary human pDCs. PBMCs were separated from peripheral blood of healthy donors, then the cells were analyzed by flow cytometry for pDCs. PBMCs were primed with 1 μM CpG-A or 1 μM CpG-B or 5 μg/mL IMQ for 2.5 h (**A**–**D**). In parallel, 20 μM nigericin was applied for 1 h to induce the NLRP3-dependent cleavage of caspase-1 and IL-1β, where it is indicated (**B**,**D**). The percentage of pro-caspase-1 (**A**), cleaved caspase-1 (**B**), pro-IL-1β (**C**) and cleaved IL-1β (**D**) -positive cells was determined within the pDC population using intracellular flow cytometry. Representative dot plots are shown in (**A**–**D**). Green dots indicate isotype controls and red dots show the specific proteins (**A**–**D**). Data are represented as means ± SD of 22 individual experiments and analyzed using one-way ANOVA followed by Bonferroni’s post-hoc test. **** *p* < 0.0001 vs. control (ctrl). IMQ: imiquimod, NIG: nigericin, SSC: side scatter.

**Figure 6 ijms-23-12154-f006:**
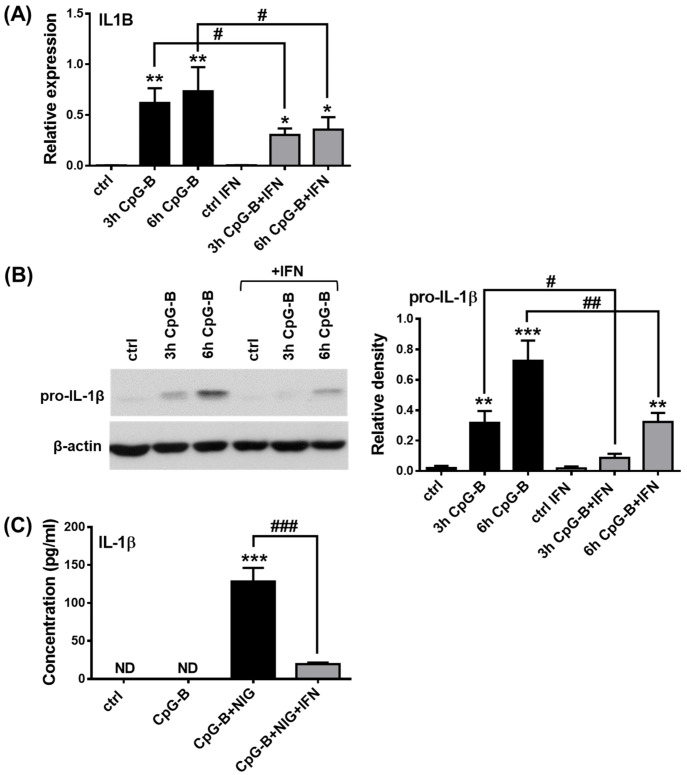
IFN-α pretreatment inhibits the IL-1β production in the human GEN2.2 pDC cell line. Cells were pretreated with 50 μg/mL recombinant IFN-α for 30 min, then activated with 1 μM CpG-B for 3 or 6 h (**A**,**C**). To induce IL-1β secretion CpG-B-treated cells were incubated in the presence of 20 μM nigericin for 4 h (**C**). The expression of *IL1B* was determined at the mRNA level by qPCR (**A**) and at the protein level by Western blotting (**B**) and ELISA (**C**). A representative blot is shown in (**B**). Bar graphs represent the mean ± SD of 6 independent experiments. Data were analyzed using one-way ANOVA followed by Bonferroni’s post-hoc test. * *p* < 0.05, ** *p* < 0.01, *** *p* < 0.001 vs. control (ctrl), # *p* < 0.05, ## *p* < 0.01, ### *p* < 0.001. IFN: interferon, ND: not determined, NIG: nigericin.

**Figure 7 ijms-23-12154-f007:**
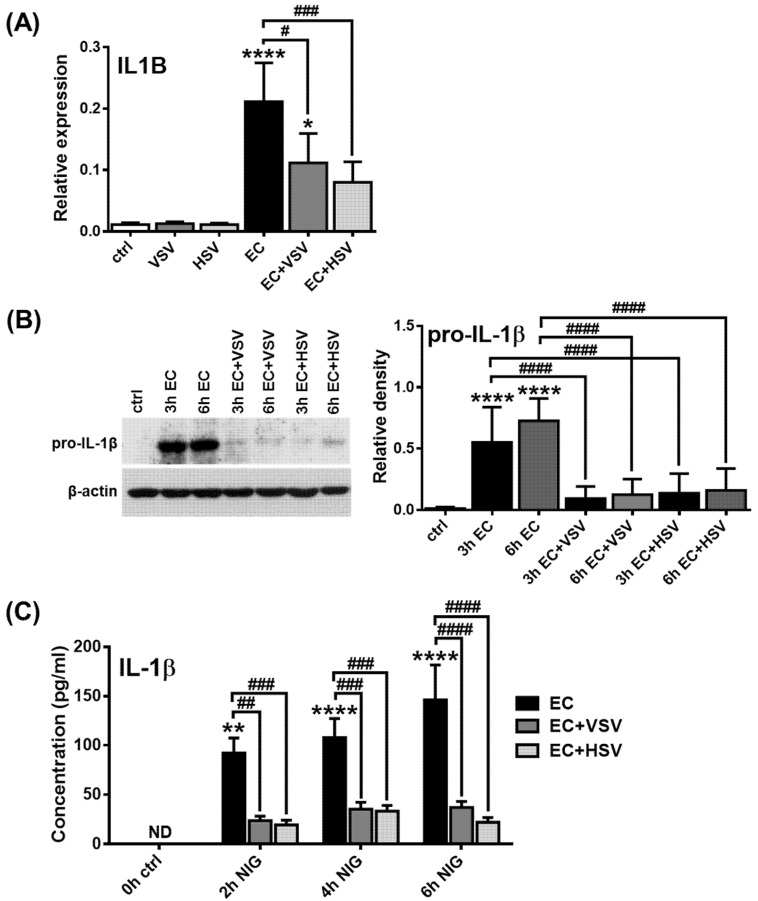
Viral pretreatment inhibits the IL-1β production in the human GEN2.2 cell line. GEN2.2 cells were incubated in the presence of VSV or HSV at a MOI of 10 for 3 h. Then cells were activated with *E. coli* (MOI 10) for 3 or 6 h (**A**–**C**). To induce IL-1β secretion *E. coli*-treated cells were incubated in the presence of 20 μM nigericin in a time-dependent manner (**C**). The expression of *IL1B* was determined at the mRNA level by qPCR (**A**) and at the protein level by Western blotting (**B**) and ELISA (**C**). A representative blot is shown in (**B**). Bar graphs represent the mean ± SD of 4 independent experiments. Data were analyzed using one-way ANOVA followed by Bonferroni’s post-hoc test. * *p* < 0.05, ** *p* < 0.01, **** *p* < 0.0001 vs. control (ctrl), # *p* < 0.05, ## *p* < 0.01, ### *p* < 0.001, #### *p* < 0.0001. VSV: *Vesicular Stomatitis Virus*, HSV: *Herpes Simplex Virus*, EC: *Escherichia coli*, NIG: nigericin, ND: not determined.

**Figure 8 ijms-23-12154-f008:**
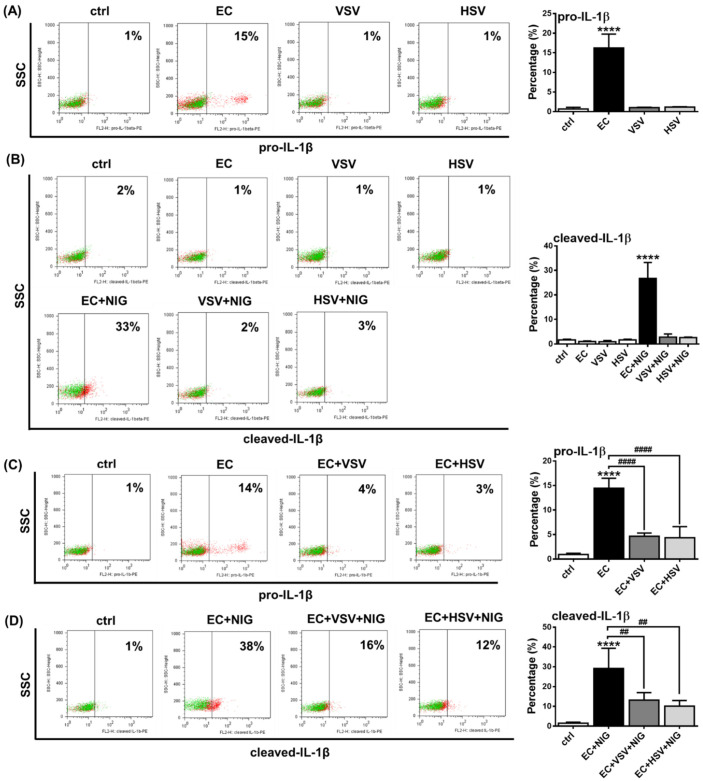
VSV or HSV pretreatment inhibits the IL-1β production in primary human pDCs. PBMCs were separated from peripheral blood of healthy individuals, then the cells were analyzed by flow cytometry for pDCs. PBMCs were incubated in the presence of E. coli, VSV or HSV at a MOI of 10 (**A**). After priming, cells were incubated in the presence of 20 μM nigericin for 1 h, where it is indicated (**B**). In parallel experiments cells were pretreated with VSV or HSV (MOI 10) for 3 h and then activated with *E. coli* (MOI 10) for an additional 3 h (**C**). To induce the cleavage of IL-1β, nigericin was added to the cells for an additional 1 h (**D**). The percentage of pro-IL-1β (**A**,**C**) and cleaved IL-1β (**B**,**D**) -positive pDCs was measured using intracellular flow cytometry. Representative dot plots are shown in (**A**–**D**). Green dots indicate isotype controls and red dots show the specific proteins (**A**–**D**). Data are represented as means ± SD of 4 individual experiments and analyzed using one-way ANOVA followed by Bonferroni’s post-hoc test. **** *p* < 0.0001 vs. control (ctrl), ## *p* < 0.01, #### *p* < 0.0001. VSV: *Vesicular Stomatitis Virus*, HSV: *Herpes Simplex Virus*, EC: *E. coli*, NIG: nigericin, SSC: side scatter.

**Figure 9 ijms-23-12154-f009:**
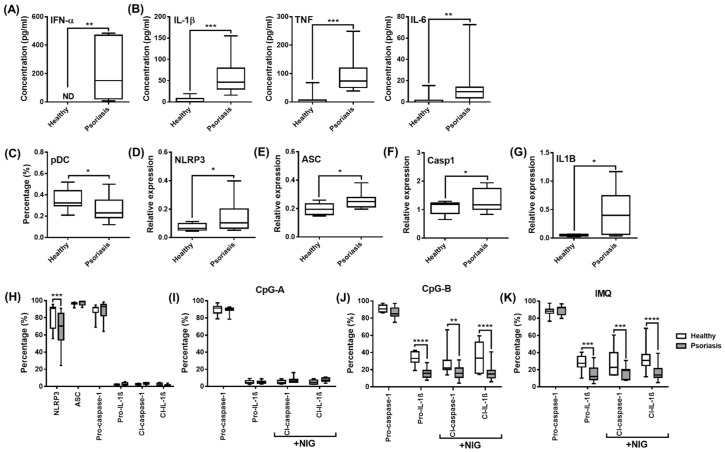
Characteristics of psoriasis patients. Peripheral blood was taken from healthy volunteers and psoriasis patients then PBMCs were separated using Ficoll-Paque gradient centrifugation and blood plasma was stored at -70 °C until analysis. The plasma concentration of IFN-α (**A**), IL-1β, TNF-α and IL-6 (**B**) was measured by ELISA. The percentage of pDCs was determined by flow cytometry (**C**). The baseline expression level of *NLRP3* (**D**)**,** *ASC* (**E**), *caspase-1* (**F**) and *IL1B* (**G**) was measured by qPCR in PBMCs. In some experiments PBMCs were left untreated (**H**) or activated with 1 μM CpG-A, 1 μM CpG-B or 5 μg/mL IMQ for 2.5 h (**I**–**K**), then stimulated with 20 μM nigericin for 1 h to induce the NLRP3-dependent cleavage of IL-1β and caspase-1 (**I**–**K**). The baseline (**H**) and the activation induced (**I–K**) expression of NLRP3 inflammasome pathway components was measured in the gated pDC population by intracellular flow cytometry. Bar graphs represent the mean ± SD of 14 independent experiments. Data were analyzed using paired Student *t* test (**A**–**F**) and two-way ANOVA followed by Bonferroni’s post hoc test (**H**–**K**). * *p* < 0.05, ** *p* < 0.01, *** *p* < 0.001, **** *p* < 0.0001 vs. healthy control. ND: not determined, IMQ: imiquimod, NIG: nigericin.

**Figure 10 ijms-23-12154-f010:**
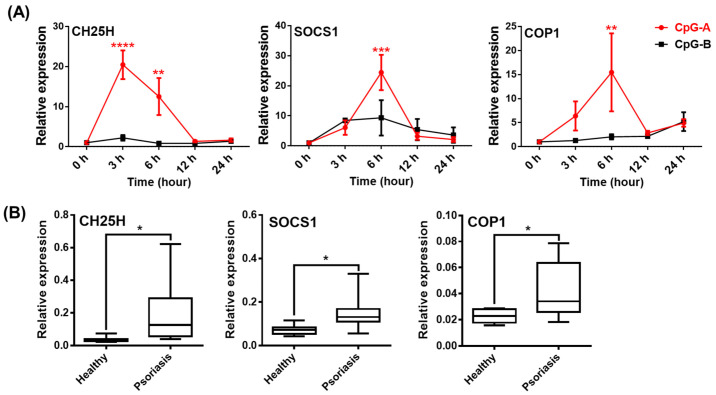
Various NLRP3 pathway inhibitors are induced by the powerful type I IFN inducer TLR9 ligands and show elevated levels in psoriasis patients. GEN2.2 cells were treated with 1 μM CpG-A or 1 μM CpG-B in a time-dependent manner and the expression of *CH25H*, *SOCS1* and *COP1* was measured by qPCR (**A**). PBMCs were isolated from the peripheral blood of healthy volunteers and psoriasis patients and the relative expression of *CH25H*, *SOCS1* and *COP1* was determined by qPCR (**B**). Bar graphs represent the mean ± SD of 4–14 independent experiments. Data were analyzed using one-way ANOVA followed by Bonferroni’s post hoc test (**A**) or paired Student *t* test (**B**). * *p* < 0.05, ** *p* < 0.01, *** *p* < 0.001, **** *p* < 0.0001 vs. control (ctrl) or healthy control.

## Data Availability

The data presented in this study are available in the article’s Figures and in Appendix A.

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
