# Peer review of "Interactions between the NLRP3-Dependent IL-1β and the Type I Interferon Pathways in Human Plasmacytoid Dendritic Cells"

_ijms, 2022, doi:10.3390/ijms232012154_

Round 1

Reviewer 1 Report

In this study, the authors demonstrated that the NLRP3-dependent IL-1β secretion pathway can be induced in pDCs under inflammatory conditions in which the type I IFN pathway is not dominant. Overall, the experiments are well performed, but I have some questions to improve the quality of the manuscript.

1)    The main issue of this paper is to analyze the interaction between the type 1 IFN pathway and the IL-1β pathway in pDC. In Figure 1-3, the expression of IL-1β by CpG-A, CpG-B, IMQ or E. coli was analyzed, but there was no result of the IFN expression. To understand the relationship with IL-1β, it is necessary to add IFN expression results.

2)    In Fig 3, MCC950, specific NLRP3 inhibitor, was treated to GEN2.2 cell line to investigate whether IL-1β secretion by pDCs is indeed NLRP3 inflammasome-dependent. It was confirmed that IL-1β expression was decreased by MCC950, but data on the inhibition of NLRP3 inflammasome were not included. The authors should confirm by western blot whether the p10 or p20 fragment of caspase-1 was reduced by MCC950. Similar to the above, caspase-1-specific inhibitor, Z-YVAD-FMK, was used in Fig S5. Although Z-YVAD-FMK significantly reduced the release of IL-1β, data on whether caspase-1 was inhibited were not included. The authors should confirm by western blot whether the p10 or p20 fragment of caspase-1 was reduced by Z-YVAD-FMK.

3)    In Figure 5, the combination of nigericin with CpG-B or IMQ increased the cleaved forms of caspase-1 and IL-1β. In order to directly show the effect of nigericin, it is necessary to add data on the cleaved form of caspase-1 and IL-1β in the absence of nigericin treatment. As mentioned above, in Figure S8, the cleaved forms of IL-1β and caspase-1 should be observed by fluorescence confocal microscopy using pDC without nigericin.

4)    In Figure 8, data for cleaved forms of IL-1β using cells not treated with nigericin should be added to indicate that neither VSV or HSV viruses induced cleavage of IL-1β.

Author Response

Response to Reviewer 1.:

We are very grateful to the Reviewer for the time and effort taken to evaluate our manuscript. Based on the Reviewer’s suggestions we made changes to the manuscript accordingly. Please read our responses below.

“1)    The main issue of this paper is to analyze the interaction between the type 1 IFN pathway and the IL-1β pathway in pDC. In Figure 1-3, the expression of IL-1β by CpG-A, CpG-B, IMQ or E. coli was analyzed, but there was no result of the IFN expression. To understand the relationship with IL-1β, it is necessary to add IFN expression results.”

We thank the Reviewer’s comment. However, we think that showing IFN production in Figure 1-3 would not be logic. To the best of our knowledge, it has not been thoroughly investigated, which primary or secondary signals are able to induce IL-1β production in human pDCs. Therefore, our primary goal was to thoroughly characterize various primary and secondary signals, which might be able to induce IL-1β production in pDCs. The aim of the Figure 1-3 was to identify those signals, which are required to induce an IL-1β response in pDCs, and thus we think that showing the IFN response of pDCs is not relevant here. However, we would like to call the Reviewer’s attention to Supplementary Figure S9 and S11, which show the kinetics of type I IFN production of pDCs upon activation with synthetic ligands (CpGA, CpG-B) and intact microbes (E.coli, VSV és HSV), respectively. Here, to measure the type I IFN levels we used the supernatants of those samples, the cell lysates of which were used to determine the levels of IL-1β by western blot in Figure 1-2.

In the manuscript we do not show the kinetics of type I IFN production upon IMQ treatments, since the GEN2.2 human pDC cell line - which we used to study the kinetics of type I IFN production due to the limiting number of primary circulating pDCs - is not able to respond to synthetic TLR7 ligands. It has been published that the synthetic ligands of TLR7 are not able to induce the translocation of IRF7 to the nucleus and thus are unable to trigger the production of type I IFNs in GEN2.2 cells (https://doi.org/10.1182/blood-2009-04-216770). Nevertheless, it must be noted that the TLR9 signaling cascade is intact in GEN2.2 cells (https://doi.org/10.4049/jimmunol.176.1.248). Based on previous publications (https://doi.org/10.1016/S0008-8749(02)00517-8, https://doi.org/10.1084/jem.20020207) the IMQ treatment can trigger type I IFN production in primary human pDCs and besides can ensure secondary signals to NLPR3 activation that we discussed in the current manuscript as well (Line 647-655). Due to these facts we studied thoroughly the regulatory effect of type I IFN production on the IL-1β responses of pDCs only in the presence of TLR9 ligands ( CpG-A, CpG-B), type I IFN response triggering viruses (VSV, HSV) and NF-kB activation inducing E.coli.

 “2)    In Fig 3, MCC950, specific NLRP3 inhibitor, was treated to GEN2.2 cell line to investigate whether IL-1β secretion by pDCs is indeed NLRP3 inflammasome-dependent. It was confirmed that IL-1β expression was decreased by MCC950, but data on the inhibition of NLRP3 inflammasome were not included. The authors should confirm by western blot whether the p10 or p20 fragment of caspase-1 was reduced by MCC950. Similar to the above, caspase-1-specific inhibitor, Z-YVAD-FMK, was used in Fig S5. Although Z-YVAD-FMK significantly reduced the release of IL-1β, data on whether caspase-1 was inhibited were not included. The authors should confirm by western blot whether the p10 or p20 fragment of caspase-1 was reduced by Z-YVAD-FMK.”

We appreciate the Reviewer constructive suggestions, based on which we inserted western blot results in Supplementary Figure S5 to show that the level of p20 fragment of cleaved-caspase-1 is reduced in the presence of MCC950 and Z-YVAD-FMK inhibitors. We also indicated the percentage of inhibition in a diagram as well. Please, find the new data in the revised version of the Supplementary Materials.

“3)    In Figure 5, the combination of nigericin with CpG-B or IMQ increased the cleaved forms of caspase-1 and IL-1β. In order to directly show the effect of nigericin, it is necessary to add data on the cleaved form of caspase-1 and IL-1β in the absence of nigericin treatment. As mentioned above, in Figure S8, the cleaved forms of IL-1β and caspase-1 should be observed by fluorescence confocal microscopy using pDC without nigericin.”

The Reviewer is absolutely right and we thank the Reviewer for calling our attention to the missing controls. In the revised version of the manuscript we inserted dot plots, which show the cleaved form of caspase-1 and IL-1β in CpG-A, CpG-B and IMQ treated cells in the absence of nigericin treatment. You can find the new data in the revised Figure 5. Furthermore, based on the Reviewer’s suggestion we also inserted microscopy images of samples without nigericin treatment in Supplementary Figure 8. Please, find the new data in the revised version of Supplementary Materials.

“4)    In Figure 8, data for cleaved forms of IL-1β using cells not treated with nigericin should be added to indicate that neither VSV or HSV viruses induced cleavage of IL-1β.”

Based on the Reviewer’s comment, we added new data to Figure 8 to indicate that VSV or HSV alone are unable to induce the cleavage of IL-1β. Please, find the new data in Figure 8 in the revised version of the manuscript.

We would like to express again our appreciation for the Reviewer’s valuable comments, which helped us to improve the quality of our paper.

Reviewer 2 Report

In this article “Interactions between the NLRP3-dependent IL-1β and the type I interferon pathways in human plasmacytoid dendritic cells” the authors propose to investigate the NLRP3 pathway components involved in the secretion of IL-1β to clarify its functional role in human pDCs and understand the effect of type I IFN in the secretion of IL-1β. With the set of experiments presented by the authors, they show the baseline levels as well as activation-induced expression levels of the different components of NLRP3-inflamasome pathway. They show that pDCs require both priming and activation to secrete the active cleaved form of IL-1β, and that human pDCs only activate the NLPR3 inflammasome pathway in response to specific stimulants. Importantly, the authors show the interaction between the type I IFNs and IL-1β pathways and how this affects the capacity of pDCs to fight pathogens. Moreover, the results present in this study can be exploited to improve treatment in certain pathologies were pDCs functions are deficient or hyperactivated. The results of the study are clearly presented, and methods are properly described. The authors also provide an excellent overview over the current literature on the topic and discuss the results of the study and its limitations. The provided conclusions are supported by the showed data.

Minor revisions are suggested:

·     -  Line 238: the authors should clarify why they perform the experiment in the presence and absence of FBS.

·      - The authors should review the manuscript since the expanded form of some abbreviations is missing.

·      - Line 282: “To our experiments…” should be “For our experiments…”

·     -  Line 1016: rewrite to become clearer: “however, the so called inflammasome-forming receptors, which excessive activation also contributes to the development of various autoimmune diseases, have not been well-characterized in pDCs so far”.

·      - The authors refer to Quantitative Real time PCR as Q-PCR and should be qPCR or in the case of the presented study since they perform reverse-transcription, RT-qPCR can also be used.

Author Response

Response to Reviewer 2.:

We are grateful for the positive evaluation of our manuscript and we thank to the Reviewer for the time and effort taken to evaluate our manuscript. Based on the Reviewer’s comments we made changes to the manuscript accordingly. Please see our responses below.

“Line 238: the authors should clarify why they perform the experiment in the presence and absence of FBS.”

We thank the Reviewer’s comment based on which we inserted a new paragraph as follows:

“We investigated IL-1β production in the absence of FBS to exclude the ATPase activity of the serum (Supplementary Figure S4A). ATP is rapidly degraded by the transmembrane ecto-ATPases. First CD39 converts ATP to AMP, which is then dephosphorylated to adenosine by CD73 (10.1007/s002100000309, 10.1007/s11302-012-9309-4). In addition, the soluble active form of both ectonucleotidases can also be found in the serum (10.3389/fimmu.2019.01729, 10.1007/s11302-012-9309-4, 10.3389/fimmu.2022.847894, 10.1016/j.jtho.2021.01.1639, 10.1186/s12967-017-1348-8) thus the hydrolysis of nucleotides can take place in the serum as well (10.1161/01.RES.65.3.531).“

“The authors should review the manuscript since the expanded form of some abbreviations is missing.”

We thank for this perception of the Reviewer. We made the appropriate changes.

“Line 282: “To our experiments…” should be “For our experiments…”

Based on the Reviewer’s comment, we rewrote the marked sentence in the revised version of the manuscript.

“Line 1016: rewrite to become clearer: “however, the so called inflammasome-forming receptors, which excessive activation also contributes to the development of various autoimmune diseases, have not been well-characterized in pDCs so far”.

We appreciate the Reviewer constructive suggestion, based on which we rephrased this sentence as follows:

„The activation of pDCs is primarily mediated by their well-known pattern recognition receptors; however, currently little is known about the expression of inflammasome-forming receptors in pDCs so far. A growing body of evidence indicates that the excessive activation of inflammasomes can also contribute to the development of various autoimmune diseases.”

“The authors refer to Quantitative Real time PCR as Q-PCR and should be qPCR or in the case of the presented study since they perform reverse-transcription, RT-qPCR can also be used.”

We thank the Reviewer’s comment, based on which we replaced Q-PCR by qPCR in the manuscript and in the Supplementary Materials as well.

We would like to express again our appreciation for the Reviewer’s valuable comments, which helped us to improve the quality of our paper

Round 2

Reviewer 1 Report

Thank you for your efforts in preparing the revised manuscript.